# MEGAN: Multi-Explanation Graph Attention Network

## Abstract

Explainable artificial intelligence (XAI) methods are expected to improve trust during human-AI interactions, provide tools for model analysis and extend human understanding of complex problems. Explanation-supervised training allows to improve explanation quality by training self-explaining XAI models on ground truth or human-generated explanations. However, existing explanation methods have limited expressiveness and interoperability due to the fact that only single explanations in form of node and edge importance are generated. To that end we propose the novel *multi-explanation* graph attention network (MEGAN). Our fully differentiable, attention-based model features multiple explanation channels, which can be chosen independently of the task specifications. We first validate our model on a synthetic graph regression dataset. We show that for the special single explanation case, our model significantly outperforms existing post-hoc and explanation-supervised baseline methods. Furthermore, we demonstrate significant advantages when using two explanations, both in quantitative explanation measures as well as in human interpretability. Finally, we demonstrate our model's capabilities on multiple real-world datasets. We find that our model produces sparse high-fidelity explanations consistent with human intuition about those tasks and at the same time matches state-of-the-art graph neural networks in predictive performance, indicating that explanations and accuracy are not necessarily a trade-off.

## 1 Introduction

Explainable AI (XAI) methods aim to provide explanations complementing a model's predictions to make it's complex inner workings more transparent to humans with the intention to improve trust and reliability, provide tools for model analysis, and comply with anti-discrimination laws (Doshi-Velez & Kim, 2017). Many explainability methods have already been proposed for graph neural networks (GNNs), as Yuan et al. (2022) demonstrate in their literature survey. However, the majority of work is focused on post-hoc XAI methods that aim to provide explanations for already existing models through external analysis procedures. In contrast to that, we demonstrate significant advantages of methods which Jiménez-Luna et al. (2020) call *self-explaining* methods. This class of models directly generates explanations alongside each prediction. One inherent advantage of many self-explaining models is their capability for explanation-supervised training. In explanation supervision the explanations are trained alongside the main prediction task to match known explanation ground truth or human-generated explanations, improving explanation quality in the process. Recently, impressive successes of explanation-supervision have been reported in the domains of image processing (Linsley et al., 2019; Qiao et al., 2018; Boyd et al., 2022) and natural language processing (Fernandes et al., 2022; Pruthi et al., 2020; Stacey et al., 2022). In the graph domain, explanation supervision is very sparsely explored yet (Gao et al., 2021; Magister et al., 2022). Inspired by the explanation-supervision successes demonstrated in other domains, especially by attention-based models, we propose our novel, self-explaining *multi-explanation graph attention network* (MEGAN) to enable effective explanation-supervised training for graph regression and classification problems.

We specifically want to emphasize our focus on graph regression tasks, which have been ignored by previous work on explanation supervision. However, we argue that graph regression problems are becoming an especially important topic due to their high relevance in chemistry and material sci-

ence applications. Typical graph XAI methods and existing work on explanation supervision provide single-channel attributional explanations, which means that each input element (nodes and edges) are associated with a single [0, 1] value to denote their importance. We argue that explanations merely indicating importance are hardly interpretable for regression tasks, as it remains unknown *for what* such explanations provide evidence: The exact predicted value, a certain value range or an especially high/low value? Due to this fact, we design our model to support an arbitrary number of explanation channels, independent of task specifications. For regression tasks, we choose 2 channels: One positive channel which indicates evidence of high target values, and one negative channel which contains evidence of low target values. We introduce a special explanation co-training routine to promote channels to behave according to those interpretations. Using synthetic and real-world datasets, we illustrate how multi-channel explanations help to improve interpretability, especially for graph regression problems.

We first validate our model on a synthetic graph regression dataset. We show that even in the single-channel case, our model outperforms existing post-hoc and explanation supervision baseline methods significantly, not only in accuracy of explanations but also in terms of prediction performance. In general, we find that our proposed model shows surprisingly good prediction performance independent from the aspect of explainability. In Appendix D we provide a benchmark comparison with numerous recent GNN architectures which shows that our model achieves state-of-the-art performance for graph regression tasks.

Moving to multi-channel explanations we show that our model creates explanations that are accurate, sparse, and faithful to predicted values. On the three real-world datasets about movie review sentiment analysis and the prediction of solubility and photophysical properties of molecular graphs, we show that our model creates explanations consistent with human intuition and knowledge. Furthermore, we show that our model reproduces known structure-property relationships for the non-trivial singlet-triplet task, supports previously hypothesized explanations, and even produces new hypotheses for explanatory motifs.

## 2 RELATED WORK

**Graph explanations.** Yuan et al. (2022) provides an overview of XAI methods that were either adopted or specifically designed for graph neural networks (GNNs). Notable ones include GradCAM (Pope et al., 2019), GraphLIME (Huang et al., 2022) and GNNExplainer (Ying et al., 2019). Jiménez-Luna et al. (2020) presents another overview of XAI methods used for the application domain of drug discovery. Sanchez-Lengeling et al. (2020) evaluate many common graph XAI methods for tasks of chemical property prediction. Henderson et al. (2021) for example introduce regularization terms to improve GradCAM-generated explanations for chemical property prediction. Most of the approaches presented here are classified as post-hoc methods, which aim to explain the decision of existing models in hindsight. Few prior works explore the class of GNNs which Jiménez-Luna et al. (2020) describe as *self-explaining*. Notably, Magister et al. (2022) introduce a self-explaining graph-concept network and Zhang et al. (2022) outline a prototype-learning approach for graphs where internal prototypes act as natural explanations.

**Explanation supervision.** During explanation-supervised training, the explanations generated by the model are trained to match a given dataset of usually human-generated explanations alongside the main prediction task. Linsley et al. (2019), Qiao et al. (2018) and Boyd et al. (2022) for example have demonstrated promising results for explanation supervision in the image processing domain. Likewise, Fernandes et al. (2022), Pruthi et al. (2022) and Stacey et al. (2022) for example demonstrate this for the language processing domain. Recently, Gao et al. (2021) introduce GNES, a method to perform GNN explanation supervision using GradCAM-generated explanations. In our work, we show that MEGAN is able to significantly improve explanation-supervision capabilities when compared to GNES. Additionally, Magister et al. (2022) emphasize that their method supports explanation supervision with human-generated explanations, however, the concept-based explanations generated by their approach are not empirically comparable to the attributional explanations produced by our model.

# 3 MULTI-EXPLANATION GRAPH ATTENTION NETWORK

## 3.1 TASK DESCRIPTION

We assume a directed graph $\mathcal{G} = (\mathcal{V}, \mathcal{E})$ is represented by a set of node indices $\mathcal{V} \subset \mathbb{N}^V$ and a set of edges $\mathcal{E} \subseteq \mathcal{V} \times \mathcal{V} \subset \mathbb{R}^E$, where a tuple $(i, j) \in \mathcal{E}$ denotes an edge from node $i$ to node $j$. Every node $i$ is associated with a vector of initial node features $\mathbf{h}_i^{(0)} \in \mathbb{R}^{N_0}$, combining into the initial node feature tensor $\mathbf{H}^{(0)} \in \mathbb{R}^{V \times N_0}$. Each edge is associated with a feature vector $\mathbf{u}_i \in \mathbb{R}^M$, combining into the edge feature tensor $\mathbf{U} \in \mathbb{R}^{E \times M}$.

We consider the problems of graph classification and graph regression, i.e. output vectors $\mathbf{y}^{\text{class}} \in \{0, 1\}^C$ (with $|\mathbf{y}^{\text{class}}| = 1$, such that the highest value of that vector identifies the class $\hat{c} = \arg\max_c \mathbf{y}_c^{\text{class}}$) or single real output values $y^{\text{reg}} \in \mathbb{R}$, respectively.

In addition, MEGAN outputs node and edge attributional explanations alongside each prediction. We define explanations as priority masks by assigning $[0, 1]$ values to each node and each edge, representing the importance of the corresponding element towards the outcome of the prediction. We generally assume that any prediction may be explained by $K$ individual importance channels, where $K$ is a hyperparameter of the model. The node explanations are given as the *node impor-tance* tensor $\mathbf{V}^{\text{im}} \in [0, 1]^{V \times K}$ and the edge explanations are given as the *edge importance* tensor $\mathbf{E}^{\text{im}} \in [0, 1]^{E \times K}$.

## 3.2 ARCHITECTURE OVERVIEW

To solve the previously defined task we propose the following *multi-explanation graph attention network* (MEGAN) architecture. Figure 1 provides a visual overview of this architecture. The net-work consists of $L$ attention layers, where the number of layers $L$ and the hidden units of each layer are hyperparameters. Each of these layers consists of $K$ individual, yet structurally identical GATv2 (Brody et al., 2022) attention heads, one for each of the $K$ expected explanation channels. Assum-ing the attention heads in the $l$-th layer have $N_l$ hidden units, then each attention head produces its own node embeddings $\mathbf{H}^{(l,k)}$, where $k \in \{1, \ldots, K\}$ is the head index. The final node embeddings $\mathbf{H}^{(l)} \in \mathbb{R}^{V \times N_l \cdot K}$ of layer $l$ are then produced by concatenating all these individual matrices along the feature dimension:

$$\mathbf{H}^{(l)} = \mathbf{H}^{(l,1)} \,||\, \mathbf{H}^{(l,2)} \,||\, \ldots \,||\, \mathbf{H}^{(l,K)} \tag{1}$$

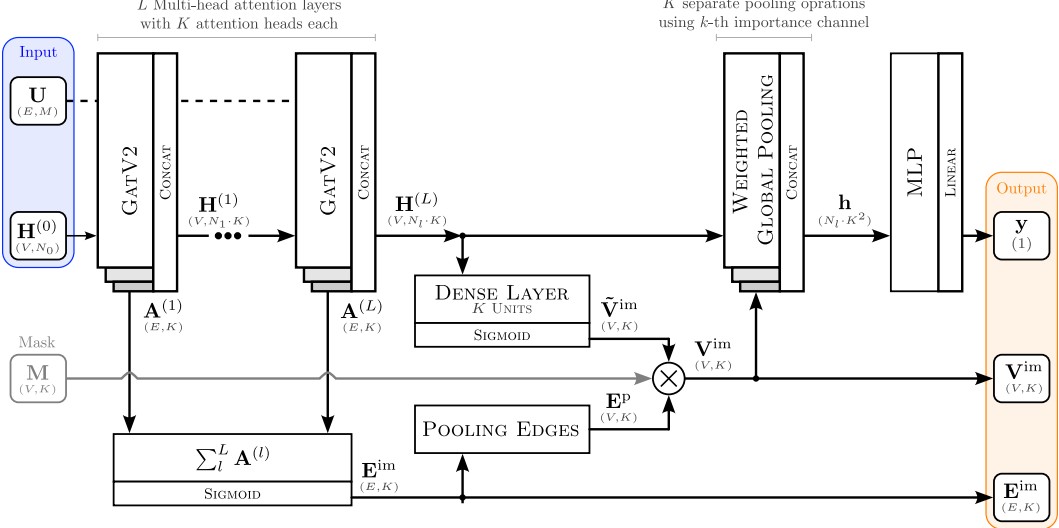

Figure 1: Multi-explanation graph attention network (MEGAN) architecture overview. Rectangle boxes represent layers; arrows indicate layer interconnections. Rounded boxes represent tensors. Intermediate tensors are also named annotated arrows. Tuples beneath variable names indicate the tensor shape, with batch dimension omitted, but implicitly assumed as the first dimension for all.

This node embedding tensor will then be used as the input to *each* of the $K$ attention heads of layer $l + 1$. Aside from the node embeddings, each attention head also produces a vector $\mathbf{A}^{(l,k)} \in \mathbb{R}^E$ of attention logits which are used to calculate the attention weights

$$\boldsymbol{\alpha}^{(l,k)} = \text{softmax}(\mathbf{A}^{(l,k)}) \tag{2}$$

of the $k$-the attention head in the $l$-th layer. The edge importance tensor $\mathbf{E}^{\text{im}} \in [0,1]^{E \times K}$ is calculated from the concatenation of these attention logit tensors in the feature dimension and summed up over the number of layers:

$$\mathbf{E}^{\text{im}} = \sigma \left( \sum_{l=1}^{L} \left( \mathbf{A}^{(l,1)} \,||\, \mathbf{A}^{(l,2)} \,||\, \ldots \,||\, \mathbf{A}^{(l,K)} \right) \right) \tag{3}$$

Based on this, a local pooling operation is used to derive the pooled edge importance tensor $\mathbf{E}^{\text{p}} \in [0,1]^{V \times K}$ for the *nodes* of the graph.

The final node embeddings $\mathbf{H}^{(L)}$ are then used as the input to a dense network, whose final layer is set to have $K$ hidden units, producing the node importance embeddings $\tilde{\mathbf{V}}^{\text{im}} \in [0,1]^{V \times K}$. The node importance tensor is then calculated as the product of those node importance embeddings $\tilde{\mathbf{V}}^{\text{im}} \in [0,1]^{V \times K}$ and the pooled edge importance tensor $\mathbf{E}^{\text{p}} \in [0,1]^{V \times K}$:

$$\mathbf{V}^{\text{im}} = \tilde{\mathbf{V}}^{\text{im}} \cdot \mathbf{E}^{\text{p}} \cdot \mathbf{M}. \tag{4}$$

The mask $\mathbf{M}$ introduced in Fig. 1 is only optionally used to compute the fidelity metric, which is introduced in Section 4.2.

At this point, the edge and node importance matrices, which represent the explanations generated by the network, are already accounted for, which leaves only the primary prediction to be explained. The first remaining step is a global sum pooling operation which turns the node embedding tensor $\mathbf{H}^{(L)}$ into a vector of global graph embeddings. For this, $K$ separate weighted global sum pooling operations are performed, one for each explanation channel. Each of these pooling operations uses the same node embeddings $\mathbf{H}^{(L)}$ as input, but a different slice $V_{:,k}^{\text{im}}$ of the node importance matrix as weights. In that way, $K$ separate graph embedding vectors

$$\mathbf{h}^{(k)} = \sum_{i=0}^{V} \left( \mathbf{H}^{(L)} \cdot \mathbf{V}_{:,k}^{\text{im}} \right)_{i,:} \tag{5}$$

are created, which are then concatenated into a single graph embedding vector

$$\mathbf{h} = \mathbf{h}^{(1)} \,||\, \mathbf{h}^{(2)} \,||\, \ldots \,||\, \mathbf{h}^{(K)} \tag{6}$$

where $\mathbf{h} \in \mathbb{R}^{N_L \cdot K^2}$. This graph embedding vector is then passed through a generic MLP whose final layer either has linear activation for graph regression or softmax activation for graph classification to create an appropriate output

$$\mathbf{y} = \text{MLP}(\mathbf{h}) \tag{7}$$

### 3.3 EXPLANATION CO-TRAINING

With the architecture as explained up to this point, there is no mechanism yet to ensure that individual explanation channels learn the appropriate explanations according to their intended interpretation (for example positive/negative evidence). We use a special explanation co-training procedure to guide explanations to develop according to pre-determined interpretations. This is illustrated in Figure 2. For this purpose, the loss function consists of two parts: The prediction and the explanation part. The explanation part is based only on the node importances produced by the network. A global sum pooling operation is used to turn the importance values of each separate channel into a single *alternate output tensor* $\hat{\mathbf{Y}} \in \mathbb{R}^{B \times K}$, where $B$ is the training batch size. This alternate output tensor is then used to solve an approximation of the original prediction problem: This can be seen as a reduction of the problem into a set of $K$ separate subgraph counting problems, where each of those only uses the subset of training batch samples that aligns with the respective channel's intended interpretation.

**Regression**    For regression, we assume $K = 2$, where the first channel represents the negative and the second channel the positive influences relative to the reference value $y_c$, which is a hyperparameter of the model and usually set as the arithmetic mean of the target value distribution in the train set. We select all samples of the current training batch lesser and greater than the reference value and use these to calculate a mean squared error (MSE) loss:

$$\mathcal{L}_{\text{exp}} = \frac{1}{2 \cdot B} \sum_{b=1}^{B} \begin{cases} (\hat{\mathbf{Y}}_{b,0} - y_c - \mathbf{Y}_b^{\text{true}})^2 & \text{if } \mathbf{Y}_b^{\text{true}} < y_c \\ (\hat{\mathbf{Y}}_{b,1} - y_c - \mathbf{Y}_b^{\text{true}})^2 & \text{if } \mathbf{Y}_b^{\text{true}} > y_c \end{cases} \tag{8}$$

**Classification**    We assume the number of channels $K = C$ is equal to the number of possible output classes $C$. We use the alternate output channel to compute an individual binary cross entropy (BCE) loss for each channel:

$$\mathcal{L}_{\text{exp}} = \mathcal{L}_{\text{BCE}}(\mathbf{Y}^{\text{class}}, \sigma(\hat{\mathbf{Y}})) \tag{9}$$

For regression as well as classification, the total loss during model training consists of these task-specific terms and an additional term for explanation sparsity:

$$\mathcal{L}_{\text{total}} = \mathcal{L}_{\text{pred}} + \gamma \mathcal{L}_{\text{exp}} + \beta \mathcal{L}_{\text{sparsity}} \tag{10}$$

where $\gamma$ and $\beta$ are hyperparameters of the training process. Explanation sparsity $\mathcal{L}_{\text{sparsity}}$ is calculated as L1 regularization over the node importance tensor. Based on this loss the gradients are calculated and the model weights are updated.

## 4    COMPUTATIONAL EXPERIMENTS

In this section, we only give a brief overview of the used datasets and experiments. Details about the datasets can be found in Appendix A. Details on hyperparameters and empirical results for experiments are listed in Appendix B

### 4.1    DATASETS

**RbMotifs.**    We create a synthetic graph regression dataset consisting of 5000 randomly generated graphs, where each node has three node features representing an RGB color value. Edges are undirected and unweighted. Some of these randomly generated colored graphs are additionally seeded with specific subgraph motifs, which consist of pre-defined color combinations and are associated with a constant value. When a motif appears in a graph, the associated value is added to that graph's overall value.

**Solubility.**    Approx. 8000 molecular graphs of the AqSolDB dataset (Sorkun et al., 2019). The target value for each graph is the value of the measured log representing the water solubility of the corresponding chemical compound. Node and edge features are generated by RDKit (Landrum, 2010).

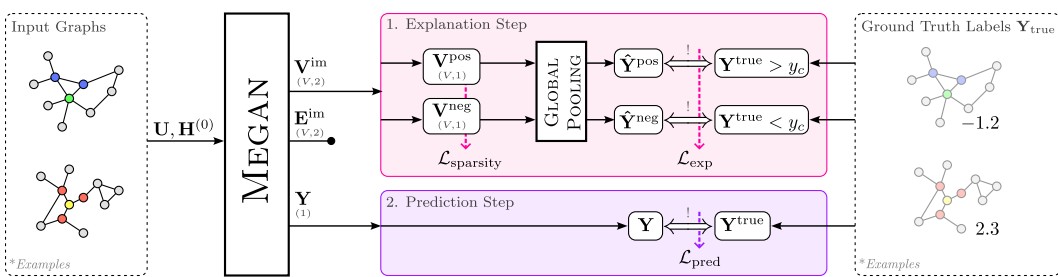

Figure 2: Illustration of the split training procedure for the regression case. The explanation-only train step attempts to find an approximate solution to the main prediction task, by using only a globally pooled node importance tensor. After the weight update for the explanation step was applied to the model, the prediction step performs another weight update based on the actual output of the model and the ground truth labels.

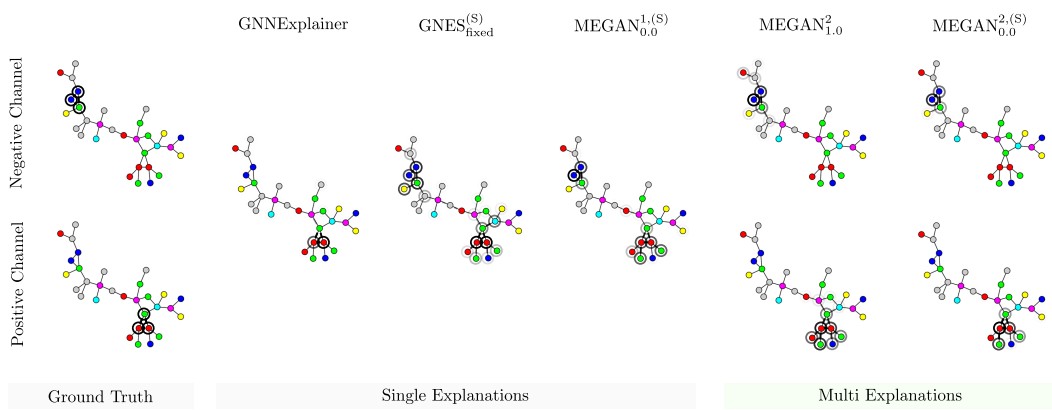

Figure 3: Examples for explanations generated for one element of the RbMotifs dataset using selected methods. We point out that in this instance GNNExplainer fails to discover the second relevant explanatory motif. Even though GNES and single-channel MEGAN correctly highlight both motifs, the single-channel explanations fail to capture the fact that both motifs represent *opposing* evidence. Only the multi-channel methods correctly capture the polarity of influence for both motifs, increasing the interpretability of the explanations.

**MovieReviews.** A dataset of 2000 movie reviews, classified by sentiments "positive" and "negative". Originally a text-based dataset from the ERASER benchmark (DeYoung et al., 2020), we pre-processed it into a graph dataset by representing each word as a node and connecting adjacent nodes by undirected and unweighted edges using a sliding window of size 2.

**TADF Singlet-Triplet Energy Splittings.** Approx. 500000 molecular graphs of the TADF dataset Gómez-Bombarelli et al. (2016). We use the singlet-triplet energy splittings $\Delta E_{ST}$ as the target value, which is one of two primary parameters to assess thermally activated delayed fluorescence (TADF) properties of molecules. Molecules with such TADF character are promising candidates for novel, low-cost OLED materials (Endo et al., 2011; Zhang et al., 2012).

### 4.2 METRICS

We measure similarity to the ground truth explanations using AUROC, as it is done by McCloskey et al. (2019). We measure sparsity on binarized explanations, which are generated by thresholding the model's continuous attributions. We also measure explanation fidelity as described in Yuan et al. (2022) and additionally provide the fidelity of random explanations for comparison. More details can be found in Appendix B.2.

For the multi-explanation models, we define the Fidelity* metric: Considering regression as an example, we argue that the positive explanation channel is faithful to the predicted output, exactly if the model output becomes significantly more *negative* when the positive channel is withheld from the model, as it is then missing all supposedly positive information about the graph. The same applies to the negative channel, which when withheld, should produce a more positive output. In this sense, we calculate a deviation $\Delta y^k$ for each channel by supplying a corresponding binary importance mask $\mathbf{M}^k$ (see Figure 1) which completely blocks channel $k$. A channel's deviation then contributes positively to the overall value if that deviation is along the expected direction as defined before:

$$\text{Fidelity*} = \frac{1}{K} \sum_{k=1}^{K} \begin{cases} +\Delta y^k & \text{if direction of deviation as expected for channel } k \\ -\Delta y^k & \text{if direction of deviation } \textit{not} \text{ as expected for channel } k \end{cases} \tag{11}$$

Consequently, positive values of Fidelity* show good alignment of explanations with their respective channel's intended interpretation, while low and negative values indicate misalignment.

Table 1: Results for computational experiments with synthetic graph regression dataset. We report the mean value for 50 independent experiment repetitions in black, as well as standard deviation in gray. We the best result for each column in bold face and underline the second best result. The first section of the table shows results for the single-explanation experiments and the second section shows the results for the dual-explanation experiments.

$^{(S)}$ Model trained explanation-supervised with ground truth explanations.
$^{(*)}$ Multi-explanation case measures Fidelity$^*$ metric introduced in Section 4.2.

| Model | MSE↓ | $r^2$ ↑ | Node AUC ↑ | Edge AUC ↑ | Sparsity ↓ | Fidelity ↑ | Fidelity$_{rand}$ |
|---|---|---|---|---|---|---|---|
| Grad × Input | $1.08_{\pm0.54}$ | $0.66_{\pm0.17}$ | $0.75_{\pm0.08}$ | $0.73_{\pm0.08}$ | $0.17_{\pm0.12}$ | $1.14_{\pm0.87}$ | $0.66_{\pm0.65}$ |
| GradCAM | $1.08_{\pm0.54}$ | $0.66_{\pm0.17}$ | $0.66_{\pm0.05}$ | $0.50_{\pm0.00}$ | $0.14_{\pm0.11}$ | $0.50_{\pm0.55}$ | $0.36_{\pm0.42}$ |
| GnnExplainer | $1.08_{\pm0.54}$ | $0.66_{\pm0.17}$ | $0.73_{\pm0.05}$ | $0.76_{\pm0.08}$ | $0.18_{\pm0.19}$ | $1.34_{\pm0.94}$ | $0.82_{\pm0.92}$ |
| GNES$_{original}^{(S)}$ | $0.91_{\pm0.04}$ | $0.71_{\pm0.01}$ | $0.59_{\pm0.01}$ | $0.58_{\pm0.02}$ | $0.12_{\pm0.07}$ | $1.11_{\pm0.66}$ | $0.51_{\pm0.48}$ |
| GNES$_{fixed}^{(S)}$ | $0.92_{\pm0.04}$ | $0.71_{\pm0.01}$ | $\underline{0.85}_{\pm0.02}$ | $\underline{0.78}_{\pm0.02}$ | $0.19_{\pm0.11}$ | $1.22_{\pm0.76}$ | $0.69_{\pm0.59}$ |
| MEGAN$_{0.0}^1$ | $\underline{0.47}_{\pm0.18}$ | $\underline{0.85}_{\pm0.06}$ | $0.74_{\pm0.12}$ | $0.70_{\pm0.08}$ | $0.14_{\pm0.08}$ | $\mathbf{3.11}_{\pm3.03}$ | $1.96_{\pm2.38}$ |
| MEGAN$_{0.0}^{1,(S)}$ | $\mathbf{0.44}_{\pm0.05}$ | $\mathbf{0.86}_{\pm0.02}$ | $\mathbf{0.97}_{\pm0.00}$ | $\mathbf{0.98}_{\pm0.00}$ | $0.17_{\pm0.11}$ | $\underline{1.44}_{\pm1.06}$ | $1.19_{\pm1.09}$ |
| MEGAN$_{1.0}^2$ | $\underline{0.27}_{\pm0.04}$ | $\underline{0.91}_{\pm0.01}$ | $\underline{0.94}_{\pm0.02}$ | $\underline{0.90}_{\pm0.06}$ | $0.10_{\pm0.06}$ | $\mathbf{2.00}^{(*)}_{\pm0.96}$ | - |
| MEGAN$_{0.0}^{2,(S)}$ | $\mathbf{0.24}_{\pm0.03}$ | $\mathbf{0.93}_{\pm0.01}$ | $\mathbf{0.99}_{\pm0.01}$ | $\mathbf{0.99}_{\pm0.01}$ | $0.09_{\pm0.06}$ | $\underline{1.91}^{(*)}_{\pm1.15}$ | - |

## 4.3 SYNTHETIC DATASET - SINGLE EXPLANATIONS

We demonstrate the capabilities of our model first for the basic single-channel explanation ($K = 1$) case. The experiment is conducted on the synthetic RbMotifs dataset. We primarily compare regression performance (MSE, $r^2$) and similarity to know ground truth explanations (Node AUC, Edge AUC). We compare with established post-hoc baselines Grad, GradCAM (Pope et al., 2019) and GNNExplainer (Ying et al., 2019). Additionally, we compare with the explanation supervision method GNES (Gao et al., 2021), which we had to modify to facilitate proper applicability to regression problems (for details refer to Appendix C). We report our results in Table 1 and illustrate examples in Figure 5.

Our results show that the explanation-supervised single-channel version MEGAN$_{0.0}^{1,(S)}$ of our network outperforms the runner-up modified GNES model by a significant margin, achieving almost perfect node and edge explanation accuracy. Additionally, our model also significantly outperforms the GCN models used as the basis for the other methods in terms of target value prediction.

## 4.4 SYNTHETIC DATASET - MULTI EXPLANATIONS

In the second experiment, we demonstrate our model's capability when moving to multi-channel explanations ($K = 2$). In this case, the existing ground truth explanations were also split into two channels, where the first channel contains all those explanatory motifs that contribute negatively to a graph's label and the second channel contains all those motifs that contribute positively. Our results firstly show that the regression performance of our model is further increased when employing an additional channel. Additionally, the explanation supervised version MEGAN$_{0.00}^{2,(S)}$ is able to correctly learn explanations, even as they are split into different channels, as indicated by the near perfect explanation accuracy. More importantly, MEGAN$_{1.00}^2$ which is not explanation-supervised but uses the explanation co-procedure achieves similarly good results.

We illustrate the added value of multi-channel explanations in the example in Figure 5. We argue that the multi-channel explanations add a highly valuable dimension to the interpretability of explanations: While some of the single-channel are able to identify the explanatory motifs accurately, the singular explanations fail to capture that in reality both of the motifs represent semantically *opposing* evidence. The multi-channel MEGAN model however is able to correctly decipher this property

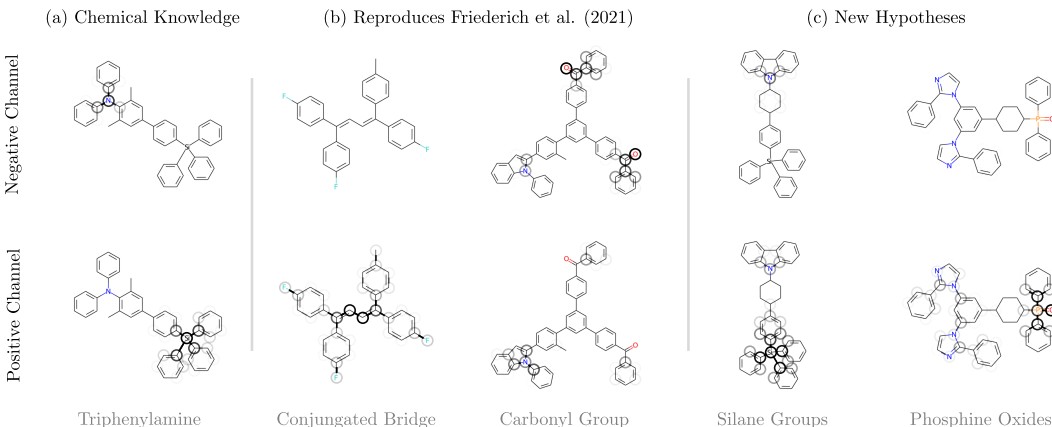

Figure 4: Selected examples for the prediction of singlet-triplet splitting energy for molecular graphs. Darker highlights for nodes and edges indicate higher predicted importance values. (a) Explanations consistent with chemical knowledge. (b) Explanations in support of hypotheses published by Friederich et al. (2021). (c) New hypotheses generated by MEGAN.

of the motifs, even without external intervention in the form of explanation supervision. Moreover, during our experiments, we notice that single-channel methods often fail to identify two motifs of opposing evidence in the same graph and we assume that this fact is a major reason for the bad performance of the baseline methods for datasets of this type.

## 4.5 REAL-WORLD DATASETS

Besides the synthetic dataset, we also apply our model on several real-world datasets. We report hyperparameters for these experiments and the detailed results in Appendix B.

The MovieReviews dataset requires the binary classification of movie review sentiment into the two classes "negative" and "positive". In terms of classification, our model performs comparably to baseline GNN approaches from literature, yet significantly worse than state-of-the-art NLP approaches (Table 6). However, we find that our model still produces explanations consistent with human intuition, as it appropriately declares negative adjectives like "bad", "unfunny" and "disappointing" as explanations for the negative class and likewise positive adjectives like "good", "remarkable" and "clever" as explanations for the positive class. We illustrate one example movie review along with our model's explanations in Table 2 and more in Appendix E.3.

Table 2: Example for movie review sentiment classification dataset. Punctuation and capitalization were removed in pre-processing. Higher node importance values are indicated by more intense color highlights.

| Negative | Positive |
|---|---|
| overall avengers endgame was a remarkable movie and a worthy culmination of the mcu up to this point there were some genuinely heartbreaking moments and breathtaking action sequences but to be honest some of the movies i had to sit through to get here were not worth it some of the early mcu movies and series leading up to this finale i found rather bland unfunny and sometimes just downright bad but this movie was one of the best movies i have seen in a while | overall avengers endgame was a remarkable movie and a worthy culmination of the mcu up to this point there were some genuinely heartbreaking moments and breathtaking action sequences but to be honest some of the movies i had to sit through to get here were not worth it some of the early mcu movies and series leading up to this finale i found rather bland unfunny and sometimes just downright bad but this movie was one of the best movies i have seen in a while |

The Solubility dataset requires the prediction of the logS value for water solubility of various chemical compounds. For this task, our model is able to achieve state-of-the-art regression performance ($r^2 = 0.91$, see Table 5) similar to results from the literature. We find that explanations of our model accurately reproduce chemical intuition about this task. Large non-polar carbon structures are provided as explanations for negative influence, while polar oxygen and nitrogen functional groups are highlighted as evidence indicative of positive influence on the target value prediction.

We also apply our model to the prediction of the singlet-triplet energy gap from the TADF dataset. Our model achieves a high predictivity of $r^2 = 0.9$ on the regression task. Additionally, the explanations of our model are able to reproduce non-trivial structure-property relationships from chemical knowledge. As shown in Figure 4(a) for example, triphenylamine bridges are known to be associated with low energy gaps, as they cause the necessary twist angles between the fragments, decoupling electron donating and electron accepting parts of a molecule to reduce the exchange interaction between the frontier orbitals which would otherwise lower the triplet state compared to the singlet state, thus preventing undesired singlet-triplet splittings. We are also able to support hypotheses published in previous work by Friederich et al. (2021), who use an interpretable decision tree method to generate explanatory hypotheses for the same task. Our model for example also finds conjugated bridges as positive evidence and carbonyl groups as negative evidence as shown in Figure 4(b). Furthermore, we are able to propose novel hypotheses for explanatory motifs: As shown in Figure 4(c), we for example observe silane groups and phosphine oxides to be consistently highlighted as evidence for high target values. In total, we propose 5 new motifs as shown in Appendix E.2.

## 5 LIMITATIONS

Despite the encouraging experimental results, there are limitations to the proposed MEGAN architecture: Firstly, there is no hard guarantee that each channel's explanations align correctly according to their predetermined interpretations. This alignment is mainly promoted through the explanation co-training, whose influence on the network is dependent on the hyperparameter $\gamma$. We observed "explanation leakage" and "explanation flipping" on rare occasions even with reasonable values of $\gamma$. In those rare cases, explanations of one channel may faintly appear in the opposite channel or a particularly disadvantageous initialization of the network causes explanations to develop in the exact opposite channel relative to their assigned interpretation. The second limitation is in the design of the explanation co-training itself, which essentially amounts to reducing the problem to a subgraph detection/counting task. While there are many important real-world applications that can be approximated as such, it still presents an important limit to the expressiveness of our model's explanations.

## 6 CONCLUSION

In this paper, we present MEGAN, the multi-explanation graph attention network architecture. Our model uses attention mechanisms to produce node and edge attributional explanations along multiple channels for graph classification and regression tasks. Our fully differentiable and self-explaining model is specifically designed to facilitate explanation supervision. On a synthetic graph regression dataset, we demonstrate that our explanation-supervised model significantly outperforms existing baseline approaches in terms of prediction performance and explanation accuracy, achieving a near-perfect similarity to ground truth explanations. Furthermore, we emphasize the importance of moving away from single-channel and towards multi-channel explanations for regression tasks. We argue that single-channel explanations for regression tasks fail to capture the reality of *opposing* evidence. An additional channel however allows to separate explanations that provide evidence for high target values and the opposing evidence for low target values. Using a special explanation co-training routine we promote the explanation channels of our model to behave according to these pre-determined interpretations. We show that this co-training is effective in producing accurate explanations, which are faithful to the model prediction and outperform the single-channel case across all metrics. At last, we apply our model to several real-world datasets and demonstrate that the produced explanations are consistent with human intuition in sentiment analysis and chemical property prediction. We can furthermore show that our model's explanations reproduce non-trivial structure-property relationships from chemical knowledge, supports previously published explanation hypotheses, and propose novel hypotheses for explanatory motifs.

## 7 REPRODUCIBILITY STATEMENT

Our model is implemented using KGCNN framework (Reiser et al., 2021) which implements graph neural networks in TensorFlow and Keras. The full source code is publicly available and can be found at `https://github.com/awa59kst120df/graph_attention_student`. In the source repository, we have aimed to provide some simple example scripts that illustrate a basic use case of our model. Additionally, we have packaged each of our experiments as separate code modules that contain the full information about the model hyperparameters and processing steps of the results. We also provide a list of the essential hyperparameters in Appendix B.

We provide additional information on the used datasets in Appendix A. For the Solubility and MovieReviews datasets, we provide references to where they are publicly available. Our own synthetic datasets and the TADF dataset are packaged publicly available at `https://github.com/awa59kst120df/visual_graph_datasets` Furthermore, we desecribe the most important pre-processing steps for the dataset, while the exact details can be found in our source code.

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

# A  DATASET DETAILS

## A.1  RBMOTIFS

RbMotifs (red & blue motifs) is a synthetic graph regression dataset. It consists of 5000 randomly generated graphs with node counts between 10 and 40. Each node is associated with three node feature values in the range $[0, 1]$, which represent RGB color values. Node colors are randomly sampled from a predefined set of 7 colors. Edges are undirected and unweighted. Additionally to the random nodes, graphs may be seeded with one of four subgraph motifs, each associated with a constant value. The target value for each graph is then calculated as the sum of all the motif-specific values of all motifs which are contained in the graph and a random component $\delta \sim \mathcal{U}(-0.5, 0.5)$:

$$y^{\text{true}} = \sum_m y^m + \delta \tag{12}$$

where $y^m$ is the constant value associated with the $m$-th motif contained in the graph. The four subgraph motifs consist of specific combinations of colored nodes, where two motifs form a similar pair, whose nodes are either dominated by red or blue nodes. This means there are two possible red motifs and two blue motifs. The red motifs are associated with a positive constant value, while the blue motifs are associated with negative values. Figure 5 illustrates this and shows some examples of graphs from the dataset. Figure 6 shows the distribution of color values as well as the distribution of how many motifs are contained within the graphs.

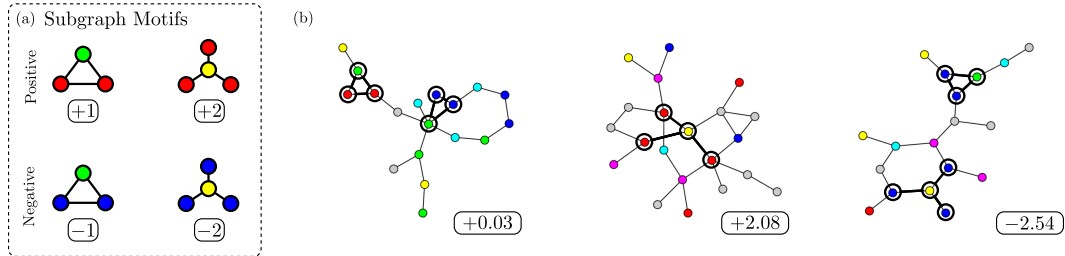

Figure 5: (a) The subgraph motifs used in the RbMotifs dataset and their associated values. (b) Example graphs from the dataset annotated with their target value

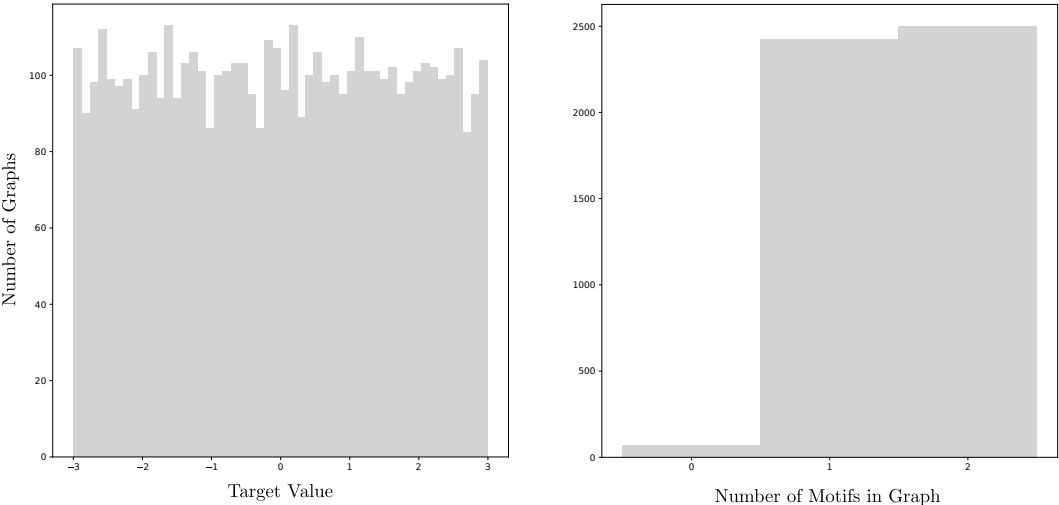

Figure 6: (a) Distribution of target values for the RbMotifs dataset. The limits $-3$ and $+3$ were imposed as hard limits during the generation process. (b) The distribution of the number of motifs contained within graphs. For 2 motifs, any combination exists.

## A.2    AQSOLDB - SOLUBILITY

We use the AqSolDB dataset introduced by Sorkun et al. (2019), which consists of 9982 chemical compounds, annotated with measurements of water solubility (logS) at room temperature. The dataset was created by merging multiple existing datasets. We follow the instructions in (Sorkun et al., 2021) and and use all of the 1290 compounds originally contained within the so-called *Huuskonen dataset* (Huuskonen, 2000) as the test set for all experiments. For the remaining elements we apply the same pre-processing steps:

- Remove all compounds which do not contain a carbon atom

- Remove adjoined mixtures

- Remove compounds that contain charged atoms

After the application of these pre-processing steps, that leaves approx. 7000 molecules in the training dataset. We process all of the SMILES strings into molecular graphs using RDKit (Landrum, 2010).

## A.3    MOVIEREVIEWS

MovieReviews is a real-world text classification dataset consisting of 2000 movie reviews from IMDB. Each movie review is to be classified into one of the two sentiment classes "positive" and "negative". Within the dataset, both classes are represented equally. We take this dataset from the ERASER benchmark (DeYoung et al., 2020) and process it in a similar way as it was done in Rathee et al. (2022) as well.

First, the strings are converted into token lists, where tokens are individual words and other sentence elements such as punctuation. In this step, we remove punctuation and capitalization. The string tokens are then converted into a 50-dimensional numeric feature vector by using a pre-trained GLOVE (Pennington et al., 2014) model. We convert the token lists into graphs by interpreting every token as a graph node and connecting each node with its neighboring nodes (according to the order of the token list/sentence order) using a sliding window of size 2. This means that every word is connected to the 2 neighboring words in both directions. Edges are undirected and unweighted. For all experiments, we use the canonical train-test split as defined by DeYoung et al. (2020).

## A.4    TADF - SINGLET TRIPLET ENERGY GAP

The dataset consists of approx. 500000 molecular graphs. Annotations were created during a high-throughput virtual screening experiment conducted by Gómez-Bombarelli et al. (2016) with the objective to discover novel materials for an application in OLED technology. Specifically, the authors aimed to discover materials that show a specific characteristic of thermally activated delayed fluorescence (TADF). This class of materials is a promising approach to avoid the high cost of typically used phosphorescent OLED materials (Endo et al., 2011; Zhang et al., 2012).
In their work, the authors use a virtual screening approach to identify particularly promising candidates. From an initial library of almost 2 million compounds, they use a neural network to predict an estimate for the delayed fluorescent rate constant ($k_{\text{TADF}}$). Candidates with especially promising values were subjected to quantum chemical simulations to obtain more accurate values. Through this process approx. 500000 compounds were annotated with the results from quantum chemical simulations. In the end, the top results were presented to human experts which selected 4 molecules that were experimentally assessed.
Along the delayed fluorescent rate constant property $k_{\text{TADF}}$, the dataset also contains annotations for the singlet-triplet gap $\Delta E_{\text{st}}$ and the oscillator strength $f$, which are the main properties from which $k_{\text{TADF}}$ is calculated.

In our work, we train our network to predict the singlet-triplet energy gap $\Delta E_{\text{st}}$ because there already exists some chemical knowledge about some structure-property relationships regarding it. Moreover, previous work by Friederich et al. (2021) already investigate possible explanatory motifs for this property using an interpretable decision tree approach.

# B  EXPERIMENT DETAILS

## B.1  EXPLANATION PRE-PROCESSING

We pre-process all explanations of MEGAN and GNNExplainer by normalizing the attribution values to a $[0, 1]$ value range. This is done w.r.t to all explanation channels, which means that the relative differences between the explanation channels remain the same.

## B.2  EVALUATION METRICS

**Explanation Accuracy.**    For cases where definite ground truth explanations are available, we measure the accuracy of the generated explanations by computing the area under the ROC curve (AUROC), as it is done in McCloskey et al. (2019), for the entire validation set. The AUROC value is in the range $[0, 1]$, where $1.0$ indicates a perfect classifier and $0.5$ indicates a random classifier.

**Sparsity.**    We calculate the sparsity as the percentage of nodes/edges contained in the binary version of the predicted node/edge importance vector. The binary version of these vectors is calculated with a threshold of $0.5$.

**Fidelity.**    For single-explanation cases, we calculate Fidelity as described in Yuan et al. (2022): The Fidelity value is the deviation of the predicted output when the given binary explanation is removed from the input of a particular sample. In this case, removal means setting all the feature values of the corresponding input elements to zero. The binary version of the explanation vectors is calculated with a simple threshold at $0.5$.

Since changes in a regressed value are harder to put into context than for a normalized classification output, which is always in the $[0, 1]$ value range, we also provide $\text{Fidelity}_{\text{rand}}$ as a reference. This is the Fidelity value that results from a randomly generated explanation mask, which has the same sparsity value as the original explanation mask. Consequently, regression explanations can be considered faithful if their Fidelity values are significantly higher than the $\text{Fidelity}_{\text{rand}}$ baseline.

## B.3  SYNTHETIC DATASET EXPERIMENTS

The first experiment on the RbMotifs dataset compares GNNExplainer with MEGAN models, which only use a single explanation channel ($K = 1$). In this experiment, the ground truth explanations are considered to consist of the union of all subgraph motifs that appear in the respective graph. The individual model and training parameters are reported in Table 3.
The GNNExplainer explanations are based on a standard multi-layer GCN (Kipf & Welling, 2017) network. The GCN network is trained on the same train set as the MEGAN models. Afterward, a GNNExplainer optimization is performed for each element of the test set to obtain the explanations. We note that we use a slightly modified implementation of GNNExplainer contained in the KGCNN library Reiser et al. (2021).
The MEGAN models used in this first experiment use only one explanation channel. Since the $K = 1$ case is not covered in Section 3.3, these models do not use any additional explanation step at all ($\gamma = 0$).

The second experiment compares different MEGAN configurations for the dual-explanation case $K = 2$. In this case, ground truth explanations are split into two channels. The first channel contains all blue (negative) motifs that appear in the graph and the second channel contains all red (positive) motifs. The individual model and training parameters are reported in Table 3.

The results of 50 independent repetitions of these experiments can be found in Table 1.

## B.4  REAL-WORLD DATASET EXPERIMENTS

Additional to the experiments for the synthetic dataset we also perform experiments with two real-world datasets: The prediction of water solubility for chemical compounds and the sentiment classification of movie reviews. For both experiments, we only report one configuration of the MEGAN architecture. The model and training parameters can be found in Table 4. For each experiment,

Table 3: Model and Training parameters used for the experiments with the synthetic dataset Rb-Motifs. The columns from left to right are: The model name, the learning rate, the batch size, the number of training epochs, the number of convolutional layers used for the network, the hidden units used for each of the convolutional layers, the hidden units used for the MLP layers, the sparsity coefficient and the total number of parameters of the model.

| Model | LR | BS | Epochs | Depth | Conv. Units | MLP Units | $\beta$ | # Param. |
|---|---|---|---|---|---|---|---|---|
| GCN - Baselines | 0.001 | 64 | 100 | 3 | $(5,5,5)$ | $(1)$ | - | 86 |
| GCN - GNES | 0.001 | 64 | 100 | 3 | $(5,5,5)$ | $(1)$ | - | 86 |
| MEGAN[1] | 0.001 | 64 | 100 | 3 | $(5,5,5)$ | $(1)$ | 1.0 | 267 |
| MEGAN[2] | 0.01 | 64 | 100 | 3 | $(5,5,5)$ | $(1)$ | 1.0 | 853 |

Table 4: Model and Training parameters used for the experiments with the real-world datasets. The columns from left to right are: The model name, the learning rate, the batch size, the number of training epochs, the number of convolutional layers used for the network, the hidden units used for each of the convolutional layers, the hidden units used for the MLP layers, the sparsity coefficient and the total number of parameters of the model. The specified dropout percentages indicate the dropout which is applied after *each* layer.

| Dataset | LR | BS | Epochs | Depth | Conv. Units | MLP Units | $\beta$ | # Param. |
|---|---|---|---|---|---|---|---|---|
| Solubility | 0.001 | 512 | 250 | 5 | $(50,50,50,50,50)$ 30% Dropout | $(50,20,1)$ 0% Dropout | 1.0 | 150593 |
| MovieR. | 0.001 | 50 | 50 | 5 | $(50,50,50,50)$ 30% Dropout | $(50,20,2)$ 0% Dropout | 1.0 | 117914 |
| TADF | 0.001 | 1024 | 25 | 5 | $(50,50,50,50,50)$ 0% Dropout | $(50,20,1)$ 0% Dropout | 1.0 | 150593 |

we briefly optimize the hyperparameters manually. Most importantly, we find that dropout regularization proves increasingly useful for increasing numbers of node features and layers. The given dropout percentages are applied after each layer.

The results of 50 independent repetitions of the solubility experiment can be found in Table 5. We also report the results of Sorkun et al. (2021) for the same test set.
Overall repetitions, our model performs consistently well in terms of predictivity ($R^2 = 0.91$), although the results are slightly worse than those achieved by the consensus model employed by Sorkun et al. (2021). However, we especially point out the high Fidelity* value for our model. On the one hand, this indicates that the explanation co-training effectively promotes the learned explanations to remain truthful to the intended interpretations of the respective channels. On the other hand, this also indicates that the explanations which are found by the model can indeed be interpreted as positive and negative influences on the concept of solubility in general.

Table 5: Results for computational experiments with solubility dataset. For our own experiments, we report the mean value for 50 independent experiment repetitions in black, as well as the standard deviation of the distribution in gray.

| Source | Model | RMSE | $R^2$ | Node Sparsity | Edge Sparsity | Fidelity* |
|---|---|---|---|---|---|---|
| Sorkun et al. (2021) | Consensus | 0.54 | 0.93 | - | - | - |
| ours | MEGAN$_{1.0}^{2}$ | $0.60_{\pm 0.02}$ | $0.91_{\pm 0.01}$ | $0.29_{\pm 0.14}$ | $0.29_{\pm 0.13}$ | $1.20_{\pm 0.34}$ |

The results of 50 independent repetitions of the MovieReviews experiment can be found in Table 6. We also report the results of DeYoung et al. (2020), who use a BERT encoder and LSTM network, and Rathee et al. (2022), who use the same pre-processing steps to provide a baseline for a simple GCN network.

Overall, DeYoung et al. (2020) clearly show the best classification performance. We believe this is due to their usage of a state-of-the-art BERT language model, which has recently proven very powerful for multiple language processing tasks. To our surprise, however, our results are marginally worse than those of a GCN baseline model reported by Rathee et al. (2022). We hypothesize that this is due to our usage of a 50-dimensional GLOVE model instead of the full 300-dimensional version used by Rathee et al. (2022). In the future, we want to investigate different language encoder models in junction with our model.

Table 6: Results for computational experiments with solubility dataset. We report the median value for 50 independent experiment repetitions in black, as well as the 75th percentile (upper) and 25th percentile (lower) of the distribution in gray.

| Source | Model | F1 | Node Sparsity | Edge Sparsity | Fidelity* |
|---|---|---|---|---|---|
| DeYoung et al. (2020) | BERT+LSTM | 0.97 | - | - | - |
| Rathee et al. (2022) | GLOVE+GCN | 0.85 | - | - | - |
| ours | $\text{MEGAN}_{1.0}^2$ | $0.84_{\pm 0.03}$ | $0.02_{\pm 0.01}$ | $0.02_{\pm 0.01}$ | $0.76_{\pm 0.17}$ |

The results of 3 independent repetitions of the TADF experiment can be found in Table 7. For this experiment, we conducted only 3 independent repetitions due to the drastically increased computation time for the larger dataset. We are not able to provide a direct comparison from the literature because the original authors Gómez-Bombarelli et al. (2016) only publish their results for the prediction of the $k_{\text{TADF}}$ property. Using a neural network they achieve $R^2 = 0.92$ for the prediction of $k_{\text{TADF}}$.

In regards to our own results, we can summarize that we are able to achieve overall good predictivity for the main prediction task as well. The network generates explanations that are sparse and faithful to the respective channel's intended interpretation, as it can be seen by the positive value of Fidelity*. One thing of note is that the Fidelity* value is much lower when compared to the solubility experiment (see Table 5). This is most likely due to the overall different value ranges of the two tasks. While the effective target value range of the solubility dataset is $[-16, 2]$ the value range for the singlet-triplet energy gap is much smaller with $[0, 3]$. Thus, deviations caused by masking individual importance channels are generally expected to have smaller absolute values.

Table 7: Results for computational experiments with TADF dataset. We report the mean value for 3 independent experiment repetitions in black, as well as standard deviation of the distribution in gray.

| Source | Model | RMSE | $R^2$ | Node Sparsity | Edge Sparsity | Fidelity* |
|---|---|---|---|---|---|---|
| ours | $\text{MEGAN}_{1.0}^2$ | $0.13_{\pm 0.00}$ | $0.90_{\pm 0.01}$ | $0.09_{\pm 0.05}$ | $0.09_{\pm 0.05}$ | $0.68_{\pm 0.38}$ |

## C  GNES IMPLEMENTATION

In Table 1 we compare the results of our own model with various baseline methods from the literature, including the GNES framework as it was proposed by Gao et al. (2021). In the GNES framework, the authors propose to use existing differentiable post-hoc explanation methods to facilitate explanation supervision. For that purpose the authors propose a generalized formulation of the explanation supervision loss, which consists of one term for node-level loss and one term for

edge-level loss:

$$\mathcal{L}_{\text{Att}}(\langle M, M' \rangle, \langle E, E' \rangle) = \alpha_n \text{dist}(M, M') + \alpha_e \text{dist}(E, E') \tag{13}$$

where $M$ and $E$ are the node and edge-level explanations generated by the model and $M'$ and $E'$ are the corresponding ground truth explanations.

The authors additionally introduce generalized formulations for node and edge-level explanations which can be derived from a model. They define the attributional explanation for node $n$ at layer $l$ w.r.t. to the predicted output for class $c$ as

$$M_n^{(l)} = || \, \text{ReLU}(g(\frac{\partial y_c}{\partial F_n^{(l)}}) \cdot h(F_n^{(l)})) \, || \tag{14}$$

where $F_n^{(l)}$ denotes the node's activation at layer l. $g(\cdot)$ and $h(\cdot)$ are generic functions that can be defined for specific implementations of explanation methods.

Similarly, the attributional explanation for an edge between nodes $n$ and $m$ at layer $l$ is defined as:

$$E_{n,m}^{(l)} = || \, \text{ReLU}(g(\frac{\partial y_c}{\partial F^{(l)}} \cdot \frac{\partial F^{(l)}}{\partial A_{n,m}}) \cdot h(F_n^{(l)}, F_m^{(l)})) \, || \tag{15}$$

where the term $\frac{\partial y_c}{\partial F^{(l)}} \cdot \frac{\partial F^{(l)}}{\partial A_{n,m}}$ represents the gradient of the edge. $g(\cdot)$ and $h(\cdot)$ are again generic functions that can be defined for specific implementations of explanation methods.

The authors show that by choosing the functions $g(\cdot)$ and $h(\cdot)$ appropriately, this formulation can be used to replicate several existing post-hoc explanation methods such as simple gradient-based saliency maps (GRAD), GradCAM and Excitation Backpropagation (EB).

For our experiments, we were not able to use the code provided at `https://github.com/YuyangGao/GNES`, as their implementation exclusively supports binary classification problems and is limited to a batch size of 1. Consequently, we re-implement GNES based on the KGCNN (Reiser et al., 2021) library to support graph regression tasks and arbitrary batch sizes. Table 1 we list two variations of GNES, which we call GNES$_{\text{original}}$ and GNES$_{\text{fixed}}$. For GNES$_{\text{original}}$ we have adapted the original formulation as previously described. We have found that this variant performs very poorly in terms of similarity to the explanation ground truth, even below the other baseline methods. For GNES$_{\text{fixed}}$ we modify the basic formulation by replacing the ReLU$(\cdot)$ operation with the absolute value operation $|| \cdot ||$ and find that this variant performs significantly better, outperforming the other baseline approaches.

Our experiments with Grad and GradCAM explanations on regression datasets show that in the presence of *opposing* evidence within the graphs, the gradients of the network w.r.t. to the regression output also tend to be positive and negative. We believe that this is the main reason for the bad performance of the GNES$_{\text{original}}$ variant. The ReLU operation essentially discards half of all evidence, thus leading to a particularly bad performance on the RbMotifs dataset, which was specifically engineered to contain opposing explanatory motifs.

## D  GNN BENCHMARK

As already pointed out in Section 4, the results in Table 1 indicate that our proposed MEGAN architecture significantly outperforms baseline methods in terms of similarity to explanation ground truth as well as prediction accuracy. All baseline explanation methods used in our experiments are based on a simple 3-layer GCN (Kipf & Welling, 2017) architecture (see Table 3). On the RbMotifs dataset, this simple architecture achieves a mean predictivity of up to $r^2 = 0.71$. In contrast to that, a 3-layer dual-channel MEGAN model achieves a mean predictivity of up to $r^2 = 0.90$ on the same dataset.

Based on these results we compare our MEGAN architecture with several state-of-the-art GNN architectures and find that our model is able to achieve state-of-the-art performance on many graph regression and classification tasks. For comparison, we use the standard implementation of several recently proposed model architectures from the KGCNN library (Reiser et al., 2021). Continuously updated benchmarking results and hyperparameter configurations can be found at `https://github.com/aimat-lab/gcnn_keras/tree/master/training/results`.

We report some benchmarking results in Table 8 and Table 9. In both cases, our model ranks second best when compared to some of the current state-of-the-art GNNs.

Table 8: Benchmarking results for the ESOL dataset (Delaney, 2004), which consists of 1128 chemical compounds and their corresponding water solubility. We report mean absolute error (MAE) and root mean squared error (RMSE). Each cell shows the mean value of random 5-fold cross-validation in black and the standard deviation in gray. Results are sorted by performance. Additionally, we highlight the best results in bold and underline the second-best results.

| Model | Epochs | MAE ↓ | RMSE ↓ |
|---|---|---|---|
| GCN | 800 | $0.59_{\pm 0.03}$ | $0.81_{\pm 0.05}$ |
| Megnet | 800 | $0.54_{\pm 0.01}$ | $0.77_{\pm 0.04}$ |
| HamNet | 400 | $0.55_{\pm 0.02}$ | $0.76_{\pm 0.02}$ |
| GraphSAGE | 500 | $0.50_{\pm 0.04}$ | $0.72_{\pm 0.08}$ |
| MAT | 400 | $0.53_{\pm 0.03}$ | $0.72_{\pm 0.04}$ |
| NMPN | 800 | $0.50_{\pm 0.02}$ | $0.71_{\pm 0.05}$ |
| GAT | 500 | $0.49_{\pm 0.02}$ | $0.70_{\pm 0.04}$ |
| GIN | 300 | $0.50_{\pm 0.02}$ | $0.70_{\pm 0.03}$ |
| CMPNN | 600 | $0.48_{\pm 0.03}$ | $0.68_{\pm 0.02}$ |
| INorp | 500 | $0.49_{\pm 0.01}$ | $0.68_{\pm 0.03}$ |
| GATv2 | 500 | $0.47_{\pm 0.03}$ | $0.67_{\pm 0.03}$ |
| Schnet | 800 | $0.46_{\pm 0.03}$ | $0.65_{\pm 0.04}$ |
| DimeNetPP | 872 | $0.46_{\pm 0.04}$ | $0.65_{\pm 0.07}$ |
| DMPNN | 300 | $0.45_{\pm 0.02}$ | $0.63_{\pm 0.02}$ |
| AttentiveFP | 200 | $0.46_{\pm 0.01}$ | $0.63_{\pm 0.03}$ |
| MEGAN | 400 | $\underline{0.44}_{\pm 0.03}$ | $\underline{0.60}_{\pm 0.05}$ |
| PAiNN | 250 | $\mathbf{0.43}_{\pm 0.02}$ | $\mathbf{0.60}_{\pm 0.02}$ |

Table 9: Benchmarking results for the Lipop dataset (Wu et al., 2018), consists of 4200 chemical compounds compounds. Graph labels for regression are octanol/water distribution coefficient (logP at pH 7.4). We report mean absolute error (MAE) and root mean squared error (RMSE). Each cell shows the mean value of random 5-fold cross-validation in black and the standard deviation in gray. Results are sorted by performance. Additionally, we highlight the best results in bold and underline the second-best results.

| Model | Epochs | MAE ↓ | RMSE ↓ |
|---|---|---|---|
| GAT | 500 | $0.50_{\pm 0.02}$ | $0.70_{\pm 0.04}$ |
| INorp | 500 | $0.46_{\pm 0.01}$ | $0.65_{\pm 0.01}$ |
| Schnet | 800 | $0.48_{\pm 0.00}$ | $0.65_{\pm 0.00}$ |
| GIN | 300 | $0.45_{\pm 0.01}$ | $0.64_{\pm 0.03}$ |
| HamNet | 400 | $0.45_{\pm 0.00}$ | $0.63_{\pm 0.01}$ |
| AttentiveFP | 200 | $0.45_{\pm 0.01}$ | $0.62_{\pm 0.01}$ |
| GATv2 | 500 | $0.41_{\pm 0.01}$ | $0.59_{\pm 0.01}$ |
| PAiNN | 250 | $0.40_{\pm 0.01}$ | $0.58_{\pm 0.03}$ |
| CMPNN | 600 | $0.41_{\pm 0.01}$ | $0.58_{\pm 0.01}$ |
| MEGAN | 400 | $\underline{0.40}_{\pm 0.01}$ | $\underline{0.56}_{\pm 0.01}$ |
| DMPNN | 300 | $\mathbf{0.38}_{\pm 0.01}$ | $\mathbf{0.55}_{\pm 0.03}$ |

# E ADDITIONAL EXAMPLES

## E.1 SOLUBILITY

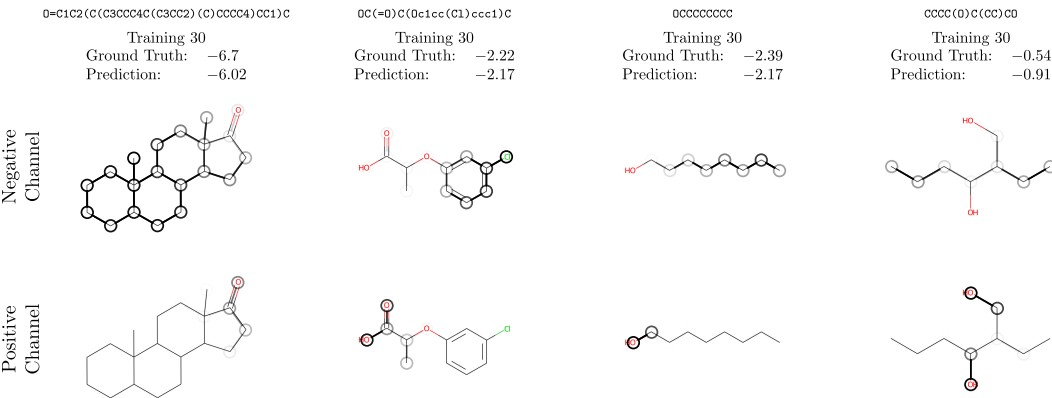

Figure 7: Examples that show that the model learns to associate carbon groups with a negative influence on the solubility value and oxygen functional groups (especially OH groups) with positive influences on the solubility value.

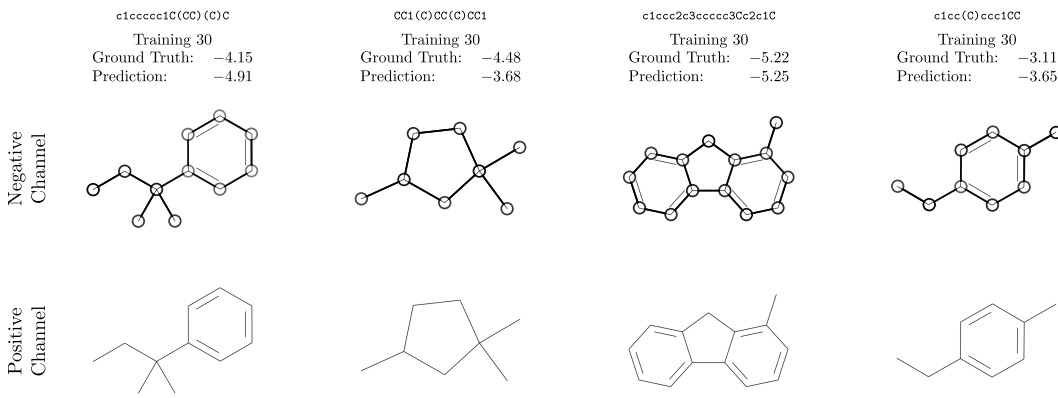

Figure 8: Examples which show that in the absence of any large carbon structures, the positive channel is usually not activated at all, further supporting the assumption that the model learns to associate carbon structures with low solubility. We point out that although the model provides a heuristical explanation consistent with chemistry knowledge here, this still shows a limitation of simple attributional explanations: Despite being explained in a similar fashion, the samples shown here still vary considerably in their actual solubility value. We argue that in such cases two simple attributional explanations as used here are not sufficient to accurately communicate the underlying reason for those differences in value.

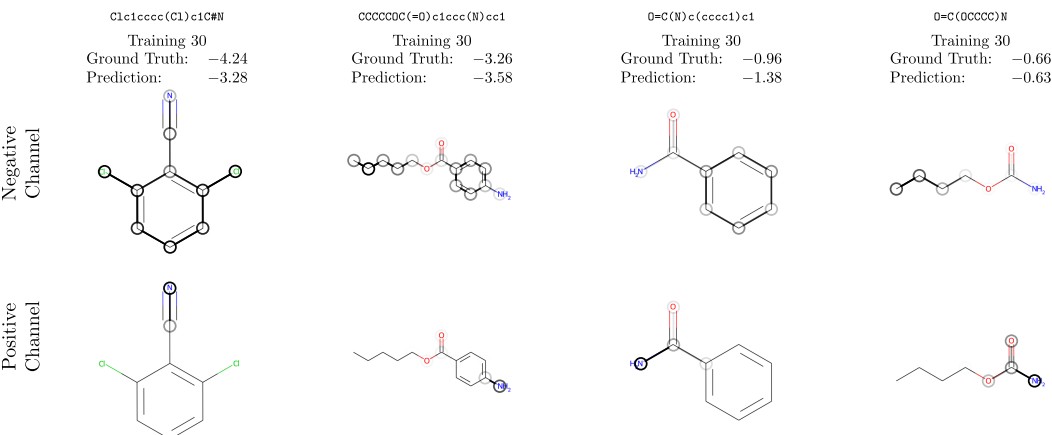

Figure 9: Examples which show that the model learns to associate nitrogen functional groups with positive influences on the solubility value as well.

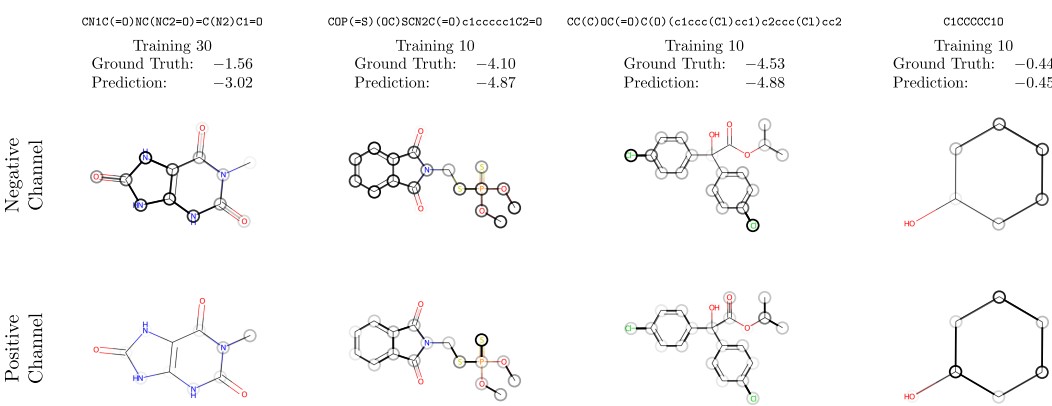

Figure 10: Examples that show that the quality of the explanations can be inconsistent within as well as in between independent repetitions of model training. The first sample is taken from the 30th repetition of the solubility experiment, from which all the good examples of the previous figures have been drawn as well. It incorrectly shows a strong activation of the negative channel and a weak activation of the positive channel, even though there are many characteristic oxygen and nitrogen functional groups present. In this case, the faulty explanation is actually reflected in the relatively large error in the model's prediction as well. The other three samples were drawn from the 10th repetition, which shows worse explanations overall. Despite the relatively accurate predictions, all three samples show very indiscriminate explanations, that feature a lot of similar activations in both channels.

### E.2 Singlet Triplet Energy Splitting

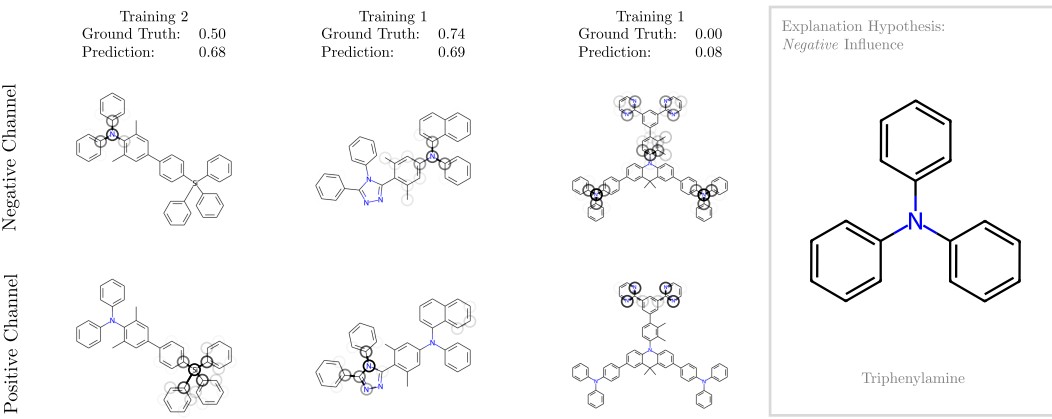

Figure 11: Selected examples, which illustrate the structure-property explanations generated by our model. We find that triarylamine bridges are consistently highlighted in the negative explanation channel as evidence for lower target values. We find these explanations consistent with chemical knowledge: "Low singlet-triplet splittings in TADF molecules are typically achieved by decoupling electron donating and accepting parts of a molecule to reduce the exchange interaction between the frontier orbitals which would otherwise lower the triplet state compared to the singlet state and open an undesired singlet-triplet splitting. The decoupling of the fragments can be achieved by introducing twist angles close to 90° between the fragments. One way to accomplish this are trialyamine bridges between the fragments" to quote Friederich et al. (2021)

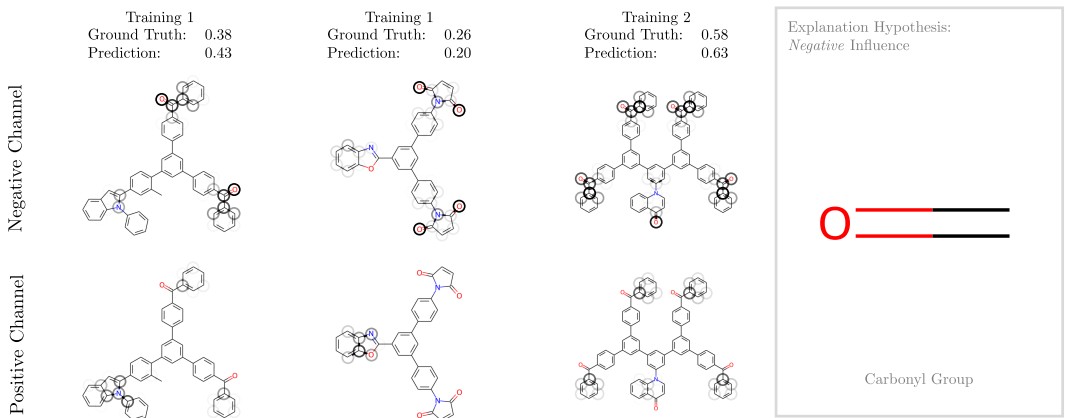

Figure 12: Selected examples, which illustrate the structure-property explanations generated by our model. We find that carbonyl groups are consistently highlighted in the negative explanation channel as evidence for lower target values. These explanations of our model directly support the hypothesis previously published by Friederich et al. (2021), who used an interpretable decision tree approach to generate their hypotheses.

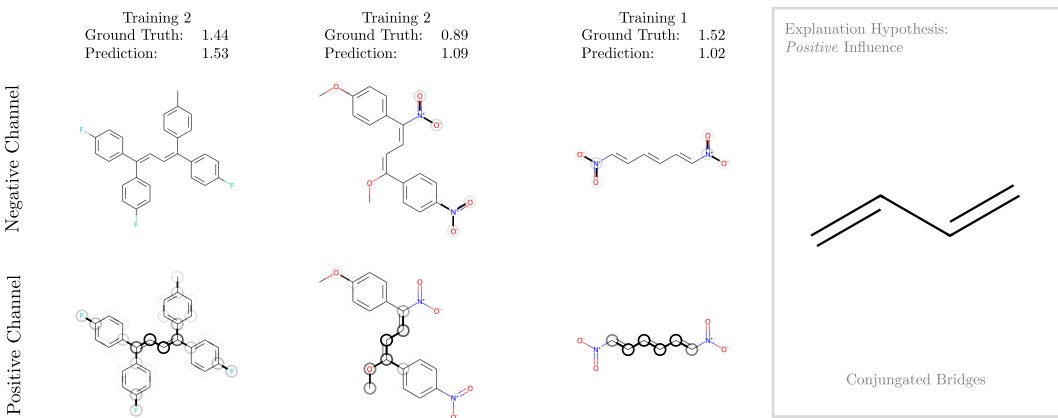

Figure 13: Selected examples, which illustrate the structure-property explanations generated by our model. We find that conjugated bridges are consistently highlighted in the positive explanation channel as evidence for higher target values. These explanations of our model directly support the hypothesis previously published by Friederich et al. (2021), who used an interpretable decision tree approach to generate their hypotheses.

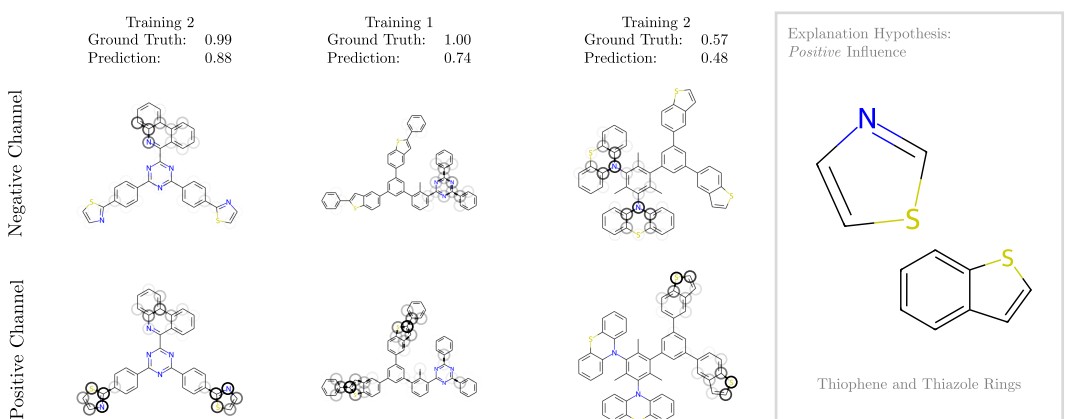

Figure 14: Selected examples, which illustrate the structure-property explanations generated by our model. We find that thiophene and thiazole rings are consistently highlighted in the positive explanation channel as evidence for higher target values. These explanations of our model directly support the hypothesis previously published by Friederich et al. (2021), who used an interpretable decision tree approach to generate their hypotheses.

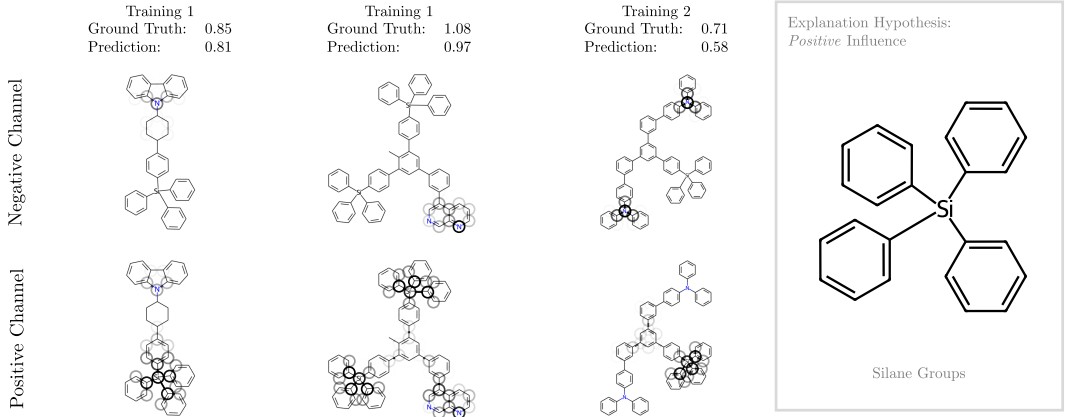

Figure 15: Selected examples, which illustrate the structure-property explanations generated by our model. We find that silane groups are consistently highlighted in the positive explanation channel as evidence for higher target values. We propose this as a new hypothesis for a possible structure-property relationship.

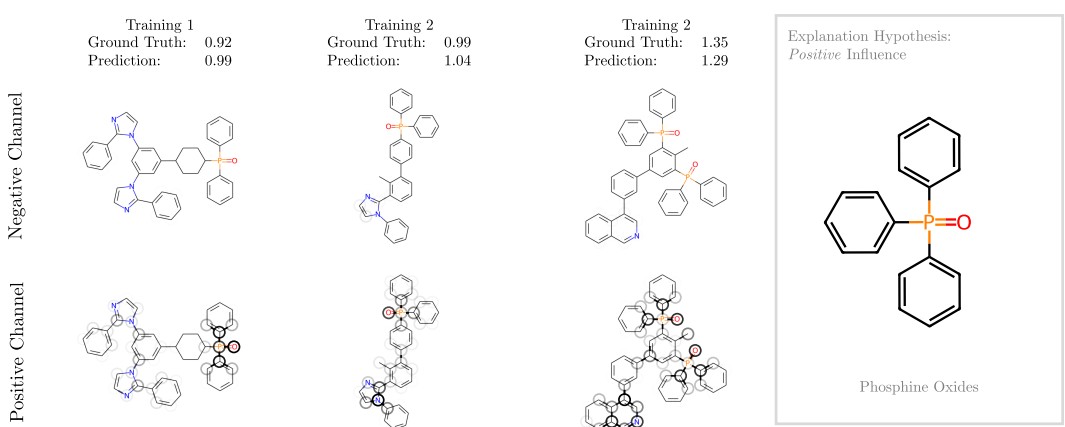

Figure 16: Selected examples, which illustrate the structure-property explanations generated by our model. We find that phosphine oxides are consistently highlighted in the positive explanation channel as evidence for higher target values. We propose this as a new hypothesis for a possible structure-property relationship.

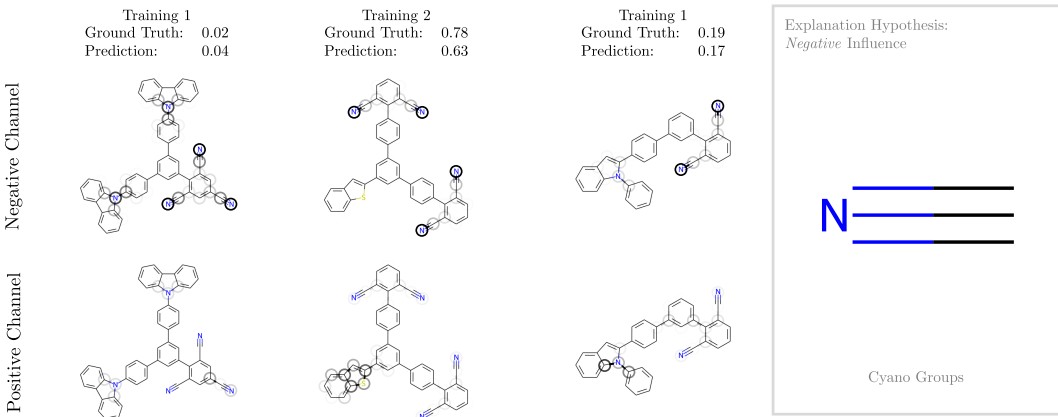

Figure 17: Selected examples, which illustrate the structure-property explanations generated by our model. We find that cyano groups are consistently highlighted in the negative explanation channel as evidence for lower target values. We propose this as a new hypothesis for a possible structure-property relationship.

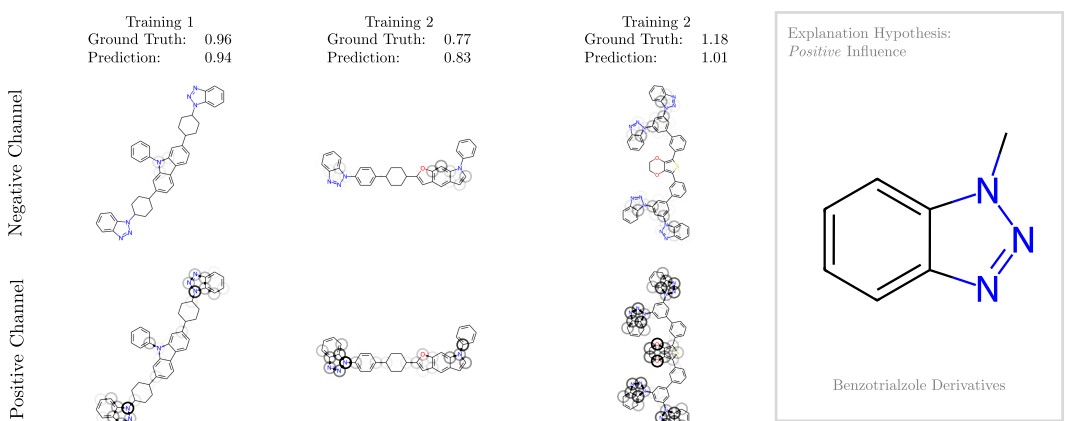

Figure 18: Selected examples, which illustrate the structure-property explanations generated by our model. We find that benzotriazole derivatives are consistently highlighted in the positive explanation channel as evidence for higher target values. We propose this as a new hypothesis for a possible structure-property relationship.

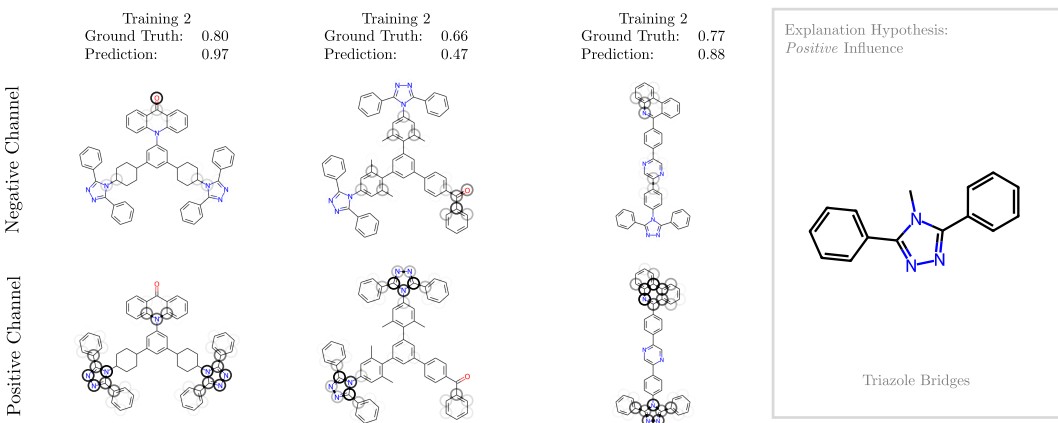

Figure 19: Selected examples, which illustrate the structure-property explanations generated by our model. We find that triazole bridges are consistently highlighted in the positive explanation channel as evidence for higher target values. We propose this as a new hypothesis for a possible structure-property relationship.

### E.3 MOVIE REVIEWS

Table 10: Example for movie review which contains positive and negative sentences, which are correctly sorted into the respective channels. However, this example also shows some sort of a bias by including the first sentence as a negative explanation. Objectively the first sentence does not contain any sentiment. The false explanation of the model is supposedly caused by the word "criminals" which is presumably often used in junction with negative adjectives.

| Negative | Positive |
|---|---|
| a couple of criminals mario van peebles and loretta devine move into a rich family house in hopes of conning them out of their jewels however someone else steals the jewels before they are able to get to them writer mario van peebles delivers a clever script with several unexpected plot twists but director mario van peebles undermines his own high points with haphazard camera work editing and pacing it felt as though the film should have been wrapping up at the hour mark but alas there was still 35 more minutes to go daniel baldwin i ca n't believe i 'm about to type this gives the best performance in the film outshining the other talented members of the cast | a couple of criminals mario van peebles and loretta devine move into a rich family house in hopes of conning them out of their jewels however someone else steals the jewels before they are able to get to them writer mario van peebles delivers a clever script with several unexpected plot twists but director mario van peebles undermines his own high points with haphazard camera work editing and pacing it felt as though the film should have been wrapping up at the hour mark but alas there was still 35 more minutes to go daniel baldwin i ca n't believe i 'm about to type this gives the best performance in the film outshining the other talented members of the cast |

Table 11: Example for an exclusively positive review. Due to the overall lack of negative adjectives, the negative channel isn't activated at all.

| Negative | Positive |
|---|---|
| this three hour movie opens up with a view of singer guitar player musician composer frank zappa rehearsing with his fellow band members all the rest displays a compilation of footage mostly from the concert at the palladium in new york city halloween 1979 other footage shows backstage foolishness and amazing clay animation by bruce bickford the performance of titties and beer played in this movie is very entertaining with drummer terry bozzio supplying the voice of the devil frank guitar solos outdo any van halen or hendrix i 've ever heard bruce bickford outlandish clay animation is that beyond belief with zooms morphings etc and actually it does n't even look like clay it looks like meat | this three hour movie opens up with a view of singer guitar player musician composer frank zappa rehearsing with his fellow band members all the rest displays a compilation of footage mostly from the concert at the palladium in new york city halloween 1979 other footage shows backstage foolishness and amazing clay animation by bruce bickford the performance of titties and beer played in this movie is very entertaining with drummer terry bozzio supplying the voice of the devil frank guitar solos outdo any van halen or hendrix i 've ever heard bruce bickford outlandish clay animation is that beyond belief with zooms morphings etc and actually it does n't even look like clay it looks like meat |

Table 12: Example which shows that the model currently doesn't understand negations and sarcasm. We point out that the partial sentence "never a bad thing" in the middle of the review is sorted into the negative channel. Another example is the first sentence: The praise it features is meant sarcastically, but it is still sorted into the positive channel.

Negative

burnt money is the perfect festival film it will show once or twice and then no one thankfully will ever have to hear from it again this film from the seattle international film festival 2001 emerging masters series is easily one of the year worst billed as a gay ' bonnie and clyde this gritty film from director marcelo pi eyro has its only highlight in a well designed title sequence two gay lovers get involved in a bank robbery that makes a gang leader whose plan they screwed up angry this causes the gang leader to send his boys out to get the gay guys one of whom may not actually be gay hiding out in a prostitute apartment the two men must fight off police and gang members in a very long showdown for the movie conclusion if caught they risk losing all the money and their love as an added emotional bonus one of the gay men is dying or something like that everything that happens is so quick and confusing i was completely lost clarity is n't exactly this movie striving virtue so it was a little hard to pick up not much could have really happened though the main events in this long two hour film are explicit homosexual and heterosexual sex graphic drug use extreme violence and strong language lots of explicit material is never a bad thing when there a reason but there no purpose to anything in this film most of the sex and violence scenes come off as silly while the heavy drug use comes off as ridiculous and depressing it appears pi eyro who co wrote with marcelo figueras from a novel by ricardo piglia purposefully adds more blood and lovemaking for his own amusement he makes the actors as sweaty and dirty as possible makes them snort cocaine gives them guns and condoms and lets them go burnt money is pointless the performances are bad it tries to thrill and shock but only causes boredom god forbid it will ever get a distributor another disappointing film from this year so called emerging masters series pass on by

Positive

burnt money is the perfect festival film it will show once or twice and then no one thankfully will ever have to hear from it again this film from the seattle international film festival 2001 emerging masters series is easily one of the year worst billed as a gay ' bonnie and clyde this gritty film from director marcelo pi eyro has its only highlight in a well designed title sequence two gay lovers get involved in a bank robbery that makes a gang leader whose plan they screwed up angry this causes the gang leader to send his boys out to get the gay guys one of whom may not actually be gay hiding out in a prostitute apartment the two men must fight off police and gang members in a very long showdown for the movie conclusion if caught they risk losing all the money and their love as an added emotional bonus one of the gay men is dying or something like that everything that happens is so quick and confusing i was completely lost clarity is n't exactly this movie striving virtue so it was a little hard to pick up not much could have really happened though the main events in this long two hour film are explicit homosexual and heterosexual sex graphic drug use extreme violence and strong language lots of explicit material is never a bad thing when there a reason but there no purpose to anything in this film most of the sex and violence scenes come off as silly while the heavy drug use comes off as ridiculous and depressing it appears pi eyro who co wrote with marcelo figueras from a novel by ricardo piglia purposefully adds more blood and lovemaking for his own amusement he makes the actors as sweaty and dirty as possible makes them snort cocaine gives them guns and condoms and lets them go burnt money is pointless the performances are bad it tries to thrill and shock but only causes boredom god forbid it will ever get a distributor another disappointing film from this year so called emerging masters series pass on by

