# OpenReview forum: "MEGAN: Multi Explanation Graph Attention Network"
_ICLR.cc/2023/Conference — Submitted to ICLR 2023_

### Official Review · Reviewer_8m6Y · 2022-10-23

**Confidence:** 3
**Correctness:** 3
**Technical Novelty And Significance:** 2
**Empirical Novelty And Significance:** 2
**Recommendation:** 3

**Clarity, Quality, Novelty And Reproducibility:**

- Clarity of the paper is good, and it is easy for readers to follow the content.
- In addition to illustrating what MEGAN is doing in each step, it will be helpful if the paper can offer more convincing insights on why MEGAN is designed this way.
- Probably there will be better reproducibility when the source codes are available.


**Strength And Weaknesses:**

- Strength
    - The paper proposes an attention-based self-explaining model for graph regression and classification, which features multiple explanation channels independent of the task specifications.
    - The paper is well organized in illustrating the overall framework and the process of how to generate multi-channel explanations. It is easy to follow the technical details.
- Weaknesses
    - The only baseline in the experiments is GNNExplainer, which was published 3 years ago, and there are quite a few new models offering better performance on explaining graph-based prediction tasks. More importantly, GNNExplainer is a post-hoc method for explaining blackbox GNN models, but MEGAN is a self-explaining models including both prediction and explanation modules. The baseline is too weak, and it is not fair to compare a post-hoc method with a self-explaining method. The paper should compare MEGAN with some more appropriate baselines, which are also self-explaining methods, such as (Gao, et al., 2021), (Zhang, et al., 2022) and (Magister et al., 2022).
        - (Gao, et al., 2021) GNES: Learning to Explain Graph Neural Networks
        - (Zhang, et al., 2022) ProtGNN: Towards Self-Explaining Graph Neural Networks
        - (Magister et al., 2022) Encoding Concepts in Graph Neural Networks
    - For evaluating the interpretability of MEGAN, quantitative analysis is only conducted on the synthetic data set. More experiments should be done on real-world datasets as well.
    - BAShapes and TreeCycles are synthetic datasets most widely used as benchmark of GNN explanation tasks. However, this paper only uses the RbMotifs dataset created by the authors themselves in the experiments. For fairness of experiments, the authors need to give an explanation for this issue as well.
- Question
    - For the explanation co-training, the reference value y_c seems to be unspecified. How do we define this value in training as it seems to affect the model's effectiveness.


**Summary Of The Paper:**

This paper proposes the multi-explanation graph attention network (MEGAN), which is an attention-based self-explaining model for graph regression and classification. MEGAN features multiple explanation channels independent of the task specifications. The edge explanations are given as the edge importance tensor, which is calculated from the concatenation of attention logit tensors. The node importance tensor, which represents node explanations, is then given as the product of node importance embeddings and the pooled edge importance tensor. The paper first evaluates MEGAN on a synthetic graph regression dataset, and further demonstrates the advantages of multi-channel explanations on two real-world datasets: the prediction of water solubility of molecular graphs and sentiment classification of movie reviews. The authors claim that MEGAN produces explanations consistent with human intuition.

**Summary Of The Review:**

The framework of MEGAN is well illustrated in this paper, but in the experiments the baseline is too weak and not appropriate.  It is not fair to compare a post-hoc method (GNNExplainer) with a self-explaining method (MEGAN).

---

> ### Author Response · Authors · 2022-11-07
> **Addressing your concerns (1/2)**
>
> Thank you very much for your constructive review. In the following comment we would like to address your concerns and outline our plan to improve the manuscript.
>
> **Baselines**
>
> Thank you for pointing out additional related work we accidentally missed! We will include those in the related work as soon as possible.
>
> We agree that other self-explaining approaches would be better baseline comparisons. In the end we were only able to implement the GNNExplainer baseline due to time constraints. For the revised version of the manuscript we will try to include a comparison to GNES, the method proposed by Gao et al.
>
> Regarding the works of Zhang et al. and Magister et al. we have come to the conclusion that although they present highly interesting approaches, it would be very difficult to compare MEGAN with these two approaches quantitatively. In our work, we aim to generate attributional explanations which assign an importance value of [0, 1] to each node/edge of the graph.
> Magister et al. on the other hand aim to provide concept-based explanations. To our understanding, the explanations they provide are of the format of logical expressions that concatenate “concepts” where each concept is a cluster of nodes / sub graph motif, such that an example explanation might be (“motif1” AND “motif2”) OR “motif3”. Such logical explanations can’t be quantitatively compared to our explanations.
> Zhang et al. use prototype learning to solve graph classification tasks. To our understanding, the explanation provided by their approach would be in the format of one or several internal prototypes which show the highest similarity for a given prediction. These prototype explanations also cannot be quantitatively compared to our attributional explanations.
>
> **BA Shapes / Tree Cycles**
>
> We agree that BA Shapes and Tree Cycles have emerged as de-facto standard datasets for Graph XAI methods. However, in contrast to the graph classification and regression datasets used in our paper, these are node classification datasets. Our MEGAN model is designed for graph-level classification and regression tasks, which are more common in domains such as chemistry and materials science. Therefore, our explanation mechanism is strongly based on the final weighted global graph pooling operation. The result of that operation is a global graph embedding vector which can then be used by an MLP to make a prediction. This procuedure is not trivially transferable to node-classification tasks.
>
> Nonetheless, we are working on an extension of the MEGAN architecture to also support node- and edge-level classification and regression tasks. However, this will most likely require some non-trivial modifications to the approach and is therefore beyond the scope of this manuscript.
>
> **Quantitative Analysis**
>
> We have conducted quantitative experiments for the used real-world datasets as well and report the results in Appendix B (due to the 9-page limit for the main body of the paper). We will refer to that in the main body of the revised version of our paper. We find that in terms of main prediction results our model performs comparable to baselines previously reported in the literature. Additionally, we can show that across multiple independent training repetitions our model consistently generates explanations that are sparse as well as faithful to the main output prediction.
>
> In Appendix C, we provide many more illustrated examples of real-world tasks. For example, we illustrate that our model correctly reproduces chemical intuition about the problem of water solubility.
> For a more details on our quantitative assessments and examples on the real-world tasks refer to our comment to Reviewer Bi9b (https://openreview.net/forum?id=H6LVUiHzYDE&noteId=fFBQjL5DCV).
>
> Additionally, for the revised version of our manuscript, we will introduce another real world dataset from material science where we can show that our model reproduces non-trivial structure-property explanations from chemical knowledge, supports previously published hypotheses and even proposes new explanatory motifs.

---

> ### Author Response · Authors · 2022-11-07
> **Addressing your concerns (2/2)**
>
> **Reproducibility**
>
> The source full source code is already anonymously published at  https://github.com/awa59kst120df/graph_attention_student. We have failed to address this through an explicit Reproducibility Statement, but we will correct this in the revised version of the manuscript.
>
> **Your Question about the Reference Value y_c**
>
> The reference value y_c is indeed an important parameter w.r.t. to the explanations. This value determines what kinds of target values are even considered “high” and which values are considered as “low”. We find that setting this value to the statistical mean of all target values of the training dataset provides good results. In the end however, it is a hyperparameter that can be determined by the user to tune what the explanations look like. We will try to make this more clear during our revision.
>
> **Planned Improvements**
>
> To summarize, for the revised version of our manuscript we plan on adding more XAI baselines for comparison. Furthermore, we plan to add comparisons to other GNN’s in terms of predictive performance for the real world tasks. Additionally, we introduce another real world dataset from material science where we can show that our model reproduces non-trivial structure-property explanations from chemical knowledge, supports previously published hypotheses and even proposes new explanatory motifs.

---

> ### Author Response · Authors · 2022-11-18
> **Manuscript Revision**
>
> Dear anonymous Reviewer,
>
> We have uploaded a revised version of our manuscript and provide a summary of the modifications in our official comment: https://openreview.net/forum?id=H6LVUiHzYDE&noteId=p_7u2cPJTr3. We hope, we were able to appropriately address your comments.

---

### Official Review · Reviewer_EFXj · 2022-10-25

**Confidence:** 3
**Correctness:** 3
**Technical Novelty And Significance:** 4
**Empirical Novelty And Significance:** 3
**Recommendation:** 6

**Clarity, Quality, Novelty And Reproducibility:**

I think the paper explains its idea clearly and the model, expermental results, and analysis are well presented. The contributioin of this paper is clear and novel. Though I didn't run the release codes, it makes me confident about it's reproducibility.

**Strength And Weaknesses:**

Strength:
1. The paper is well written and well organized. The motivation of multi-explanation is explained clearly. It's not hard to follow how the authors designes the model to have multiple explanability channel in the methodology part.
2. The idea of multi-explanation is novel and fits the graph idea well. The authors did a great job on analyzing potential issues of multi-channel  explanations and proposing a novel solution (co-training) to address some of the issues. In section 5, the authors thoroughly discussed the limitations of the proposed model.
3. The expermental results look promising. The authors showed that the explanations generated by the propose model were reasonable and aligned with the ground truth. It also outperforms the baseline model on many aspects.
4. The authors released their codes anonymously and included detailed comments and docs about codes itself and how to re-run the experiments.

Questions:
1. If you train the model twice with same datasets but different random seeds, do the same explanation channels (by index) in each run align with each other? I think this is an important aspect to discuss when you have multiple explanation channels. If it's stable, the same channel (by index) should refer to the same underlying latent space. If it's not stable, is there a way we can make it stable, or we can somehow find a mapping between new channels and old channels? For example, someone uses this model and spends quite a lot of resources to figure out what each explanation channel roughly means, then they get some new graphs data and need to re-train the model, how can they make use of the understandings of old explanation channels? Would like to see some discussions on this.
2. In the movie review datasets, the authors mentioned "our model’s accuracy is consistent with that achieved by other GNNS
as reported by ...". It would be nice to include the metrics somewhere. Maybe I missed it, but I couldn't find them in the main body or the appendix.
3. Would it be possible to run MEGAN over non graph datasets? Namely just graphs with zero edges. Just curious how MEGAN would perform comparing with those non-graph based explanability models.

**Summary Of The Paper:**

This paper introduces a new explainable graph attention network model. The highlight of this model is its multi-channel explanability. Along with outputing node predictions, the model also generates extra edges and nodes importance scores across preset channels. To along these importance scores with multi-explanation channels, the authors introduce a new way of training the model, named explanation co-training. It adds an explanation step, where the model will be penalized when the importance scores are not aligned with the ground truth label. The authors evaludated their model in one synthetic dataset and two real-world datasets and showed the proposed model gets great empirical results as well as reasonable explanations.

**Summary Of The Review:**

I think this paper is well organized and the idea will be beneficial to other researchers in this field.



---- update after seeing other reviewers' comments and authors' response
Overall I still think this is a good paper. Other reviewers called out that comparisons with many other GNN explanable models are missing and the experimental sections could've been more thorough. I think their points made sense, so I'm lowering my score to marginal accept for now.

---

> ### Author Response · Authors · 2022-11-07
> **Answering your questions**
>
> Thank you for your review, your confidence and your interest in our work.
>
> We will try to answer your questions in the following comment:
>
> **1.Question**
>
> Generally, if one would train a GNN independently on different seeds, then explanations would appear in different explanation channels (by index). This is because by default there is no mechanism to enforce any kind of pre-determined allocation of explanations to specific channels. For that reason, we introduced the explanation co-training procedure. In the co-training procedure, we explicitly assign an interpretation (e.g. positive/negative influence) to each channel prior to training, to obtain consistent explanations and reproducible channel assignments.
>
> It is a very good question how it could be possible to stabilize the explanations without the usage of the co-training. We have two basic ideas of how to approach a situation as you described:
>
> 1. An exclusivity based approach. Say we configure the model to work on a prediction task and produce 5 explanation channels. Then one option would be to add an explanation exclusivity regularization term, which would enforce all explanation channels to produce mutually exclusive explanatory motifs. We make the assumption that even for different dataset splits the same motifs would then appear in their own channels, but simply with a permutation of indices. Comparison of the old model and the new model then generates a similarity matrix based on pairwise similarities between explanations. The maximum similarity rating in each row of such a matrix should determine what the permutations of the explanations between the two models are.
>
> 2.  An explanation supervision approach. In this case, the user would predetermine the kinds of explanations associated with each channel. For example, we could algorithmically create annotations such that the “ground truth” explanations for channel 1 are of “type 1”, explanations for channel 2 are of “type 2“, etc. With these generic explanations, the network could be trained in an explanation-supervised manner such that the corresponding channels will always produce these explanations. Then the analysis for the impact of the explanations could be done in reverse: Since we now know the exact meaning of each channel we simply have to analyze what the contribution of each channel is towards the final prediction value. This could, for example, lead to the discovery that a “type x” channel contributes on average a value of say -p to the final output value.
>
> **2.Question**
>
> The results are listed in Appendix B.4 / Table 6. This table shows the classification F1 score of our model compared to the results reported by two other papers. Our results are below those achieved by DeYoung et al. as they used a state-of-the-art fine-tuned BERT language model. However, our results are comparable to Rathee et al. who have also used a graph-based approach. Generally, we think that these results show that the graph-based approach to this natural language model is still generally inferior to the specifically tailored NLP approaches. This is also reflected in the additional examples in Appendix C, which show that our model generally is able to highlight positive and negative adjectives appropriately but is not able to detect language constructs such as negations and sarcasm.
>
> **3.Question**
>
> Completely non-graph datasets (with no edges) would not be possible. This is because the node importances also partially rely on the edge importances, which would be zero for no edges. This would mean that all explanation masks would be purely zero and thus also the final prediction of the network.
>
> However, it would be possible to work on different datasets by converting those into some sort of graph first. We did this for example with the movie reviews dataset by simply assuming a connecting edge between adjecent words within a sentence. The same could be done for image data for example: By interpreting every pixel as a node and making some sort of assumption about how adjacent pixels would be connected by edges an image could be converted to a graph.
>
> Besides data from the image and natural language domain, were there any special datasets you had in mind for this?

---

> ### Author Response · Authors · 2022-11-18
> **Manuscript Revisions**
>
> Dear anonymous Reviewer,
>
> We have uploaded a revised version of our manuscript and provide a summary of the modifications in our official comment: https://openreview.net/forum?id=H6LVUiHzYDE&noteId=p_7u2cPJTr3. We hope, we were able to appropriately address your comments.

---

### Official Review · Reviewer_Bi9b · 2022-10-29

**Confidence:** 3
**Clarity, Quality, Novelty And Reproducibility:** The novelty and contributions of this…
**Correctness:** 3
**Technical Novelty And Significance:** 2
**Empirical Novelty And Significance:** 2
**Recommendation:** 3

**Strength And Weaknesses:**

Strength:
1. The research problem is very important and this paper provides different aspects to provide explanations in GNNs.
2. This paper is easy to follow and clearly written.


Weaknesses

There are some concerns regarding this paper.


1. The novelty of this paper is limited. The proposed framework of multi-explanation graph attention network is straightforward by using attention mechanisms to produce node and edge attribution explanations along multiple channels for graph classification and regression tasks.
It would be better if they detail more their novelty. Also, it's unclear the contributions of this work.

2. This work misses some advanced related works and baselines in their experiments. This work mainly compared with GNNExplainers, and this work can be more solid if they can compare some advanced GNN explanations methods.


3. Evaluation in GNNs explanations are not convincing enough.  Maybe more real-world datasets and evaluation metrics can enhance the effectiveness of the proposed method.


**Summary Of The Paper:**


This paper introduces  a multi-explanation graph attention network architecture by useing attention mechanisms to produce node and edge attribution explanations along multiple channels for graph classification and regression tasks. Experiments demonstrate the effectiveness of the proposed model.


**Summary Of The Review:**

Strength:
1. The research problem is very important and this paper provides different aspects to provide explanations in GNNs.
2. This paper is easy to follow and clearly written.


Weaknesses

There are some concerns regarding this paper.


1. The novelty of this paper is limited. The proposed framework of multi-explanation graph attention network is straightforward by using attention mechanisms to produce node and edge attribution explanations along multiple channels for graph classification and regression tasks.
It would be better if they detail more their novelty. Also, it's unclear the contributions of this work.

2. This work misses some advanced related works and baselines in their experiments. This work mainly compared with GNNExplainers, and this work can be more solid if they can compare some advanced GNN explanations methods.


3. Evaluation in GNNs explanations are not convincing enough.  Maybe more real-world datasets and evaluation metrics can enhance the effectiveness of the proposed method.

---

> ### Author Response · Authors · 2022-11-07
> **Addressing your concerns (1/2)**
>
> Thank you for your clear and constructive review. We will address all three main points of concern in our revised manuscript. Specifically, we will include additional real-world datasets and refer to additional baselines. Most importantly, we would like to convince you of the novelty of our work, which lies not in the use of graph attention layers to produce multiple explanation channels, but rather in linking them strongly to the actual predictions of the model. In that way, we are able to generate intuitive explanations (specifically for regression tasks), which are directly linked to high or low predictions and thus provide critical new insight, which cannot be provided by previous explanation models.
> We have tried to address your concerns in more detail in the following comments:
>
> **Novelty**
>
> We realize that we have not communicated the novelty and contribution of our approach clearly, so we will elaborate some more, also in the revised version of the paper. The best starting point for this is the motivation behind this development. Our major motivation was the desire to perform explanation supervision. In this process, the network is not only trained with the ground truth prediction labels but also with some explanatory annotations. Since ground truth explanations are not available for real-world tasks, these explanatory annotations are usually collected from human experts. Recently there have been successes for using explanation supervision in image processing (Linseley et al. https://openreview.net/forum?id=BJgLg3R9KQ) and natural language processing (Pruthi et al. http://arxiv.org/abs/2204.10810). Our original desire is to establish this possibility for graph neural networks as well.
>
> In those domains, explanation supervision is usually implemented through attention mechanisms. We find that in the domain of graph neural networks, graph regression problems are especially relevant due to their importance in chemistry and material science. However we argue that simple single-channel attributional explanations, which assign a single importance value in the [0, 1] range to each node/edge are insufficient to explain regression results: If a model produces a certain output value “y” and simply marks a certain motif as important, what does that motif actually explain? Does it explain that very specific value, an exceptionally high/low value or is it indicative of some value range? If we don’t know that then single attributional explanations cannot be interpreted by humans accordingly. That is why we wanted to introduce multiple channels, where the elements (nodes/edges) of the graph are not only explained by a single [0,1] range importance value but instead multiple ones. To our knowledge, we are the first to propose such a multi-channel explanation architecture for graph neural networks.
>
> Additionally, we introduce a special architectural interconnection of edge- and node-level explanations: Our edge explanations are created by aggregating the attention logits from multiple layers of multi-head GATv2 attention layers. The node-level importance masks are then in part created by an additional dense layer that acts on the learned node embeddings. These values are then multiplied by the pooled edge explanations to create the final node-level explanations. These node explanation values are then used as the weights in a global weighted pooling operation. We find that it is this semantic combination of node and edge explanations that produces superior performance and creates explanations that consist of connected sub-graph motifs rather than individual nodes and edges.
>
> Finally, we introduce an explanation co-training procedure where pre-determined interpretations for certain explanation channels are enforced in a soft manner. During the training process, the network mainly learns to predict the output values by means of the final graph embeddings. Additionally, in the explanation co-training step the network also attempts to learn the target values, by using only the explanations. This means, for the regression task, negative/low values have to be predicted simply from the presence of importance annotations in the “negative” explanation channel. Accordingly, positive/high target values have to be predicted simply from the presence of importance annotations in the “high” explanation channel.
>
> To our knowledge, we are the first to propose a self-explaining GNN architecture, which directly uses explanations as an integral part in the prediction process. We introduce an interconnection between node and edge level predictions, which we find to improve performance of the network. Additionally, explanations generated by our network are fully differentiable, through which explanation supervised training can be applied very effectively as we have demonstrated

---

> ### Author Response · Authors · 2022-11-07
> **Addressing your concerns (2/2)**
>
> **Baselines from related work**
>
> Your concern in this regard is fully justified and we acknowledge this as a weakness of our work. In fact, we would like to run a more extensive comparison with other state-of-the-art graph XAI methods. For the revised version of the manuscript, we aim to include more XAI baseline procedures to make the capabilities of our model more convincing. Additionally we aim to include comparisons to state-of-the-art GNN’s on the real world datasets as well, as we find that our model shows surprisingly good prediction performance as well.
>
> **Evaluations**
>
> In Appendix B, we provide numerical metrics for the performance of our network for the mentioned real-world tasks for solubility and movie reviews. We can show that our model shows main task prediction performance comparable to previously published baselines for each task. In terms of explanations, we can show that our network produces sparse explanations which are additionally very faithful to the primary predictions made by the network. Hereby we refer to Section 4.2 which shows that high positive values of Fidelity* indicate that each channel’s highlighted explanations indeed influence the main prediction value according to their intended interpretation. This means that the explanatory motifs in the “positive” channel of a regression network indeed contribute positively to the final result and vice versa.
>
> In Appendix C, we go beyond quantifiable metrics and provide more illustrated examples for real-world tasks. We can first show that our network learns the commonly known chemical knowledge that large carbon structures are generally associated with low solubility values (negative explanation channel) and that nitrogen and oxygen functional groups are generally associated with high solubility (positive explanation channel). For movie reviews, our network appropriately highlights sentences with negative sentiments and adjectives in the negative explanation channel and positive adjectives accordingly in the positive explanation channel.
>
> Finally, we have conducted additional experiments on a large dataset of molecular graphs where the task is to predict the singlet-triplet splittings energy (Gómez-Bombarelli et al., Nature Materials 2016, DOI: 10.1038/nmat4717) of molecules for OLED displays. We find that for this task our network is also able to reproduce chemical knowledge, support previously published hypotheses for explanatory motifs (Friederich et al., Machine Learning: Science and Technology 2021, DOI: 10.1088/2632-2153/abda08) and is able to find new explanation hypotheses. We will include these results in the revised version.
>
> **Planned Improvements**
>
> To summarize, for the revised version of our manuscript we plan on adding more XAI baselines for comparison. Furthermore, we plan to add comparisons to other GNN’s in terms of predictive performance for the real world tasks. Additionally, we introduce another real world dataset from material science where we can show that our model reproduces non-trivial structure-property explanations from chemical knowledge, supports previously published hypotheses and even proposes new explanatory motifs.

---

> ### Author Response · Authors · 2022-11-18
> **Manuscript Revision**
>
> Dear anonymous Reviewer,
>
> We have uploaded a revised version of our manuscript and provide a summary of the modifications in our official comment: https://openreview.net/forum?id=H6LVUiHzYDE&noteId=p_7u2cPJTr3. We hope, we were able to appropriately address your comments.

---

### Official Review · Reviewer_NxCN · 2022-11-04

**Confidence:** 3
**Correctness:** 3
**Technical Novelty And Significance:** 2
**Empirical Novelty And Significance:** 2
**Recommendation:** 3

**Clarity, Quality, Novelty And Reproducibility:**

The paper is well written, easy to read. The proposed architecture sues co-training which is novel but overall appears to be incremental in contribution. Hyperparameters for training procedure and architecture details for each of the experiments are needed in more detail for reproducibility.

**Strength And Weaknesses:**

Strengths:
The paper is well written and the problem is well motivated.
The proposed architecture is outlined is good detail.
Experiments are run on real world and synthetic datasets.

Weaknesses:
The notions of channels for explanations and how explanations are defined needs to be described in greater detail.
Experiments report performance at various percentiles of metrics, but statistical significance is missing.
Proposed approach is compared only with a single baseline approach, need more approaches to be covered in prior work and compared with.

**Summary Of The Paper:**

This work proposes MEGAN, a multi-explanation graph attention network model that leverages attention mechanism to produce node and edge attribute explanations along multiple channels for graph classification and prediction tasks. The proposed approach is explanation co-training: along with the prediction task, and explanation task is trained where, based on node importances, an importance value is generated per channel. Channels are defined based on the task: for regression, the explanation task is whether the the importance value is greater than y_channel, or less than y_channel, whereas for classification, the number of channels is the number of possible output classes C.

The models are compared with GNNExplainer model over datasets RbMotifs, Solubility and MovieReviews datasets. The metric used is Area under ROC curve (AUROC), which measures similarity to ground truth explanations. Another metric used is fidelity metric, which looks at faithfulness to predicted output. Experiments are also run on synthetic datasets. MEGAN reports better R^2, node and EDGE AUC and Fidelity metrics and lower MSE.

**Summary Of The Review:**

While the problem is well motivated and the paper is well written, with experiments on several real and synthetic datasets, the notion of channels for explanations seems rather limited and needs to be better described. The performance improvements over metrics for explanations needs statistical significance to be reported.

---

> ### Author Response · Authors · 2022-11-07
> **Addressing your concerns**
>
> Thank you very much for your constructive review. We would like to convince you of the novelty we see in our approach, which primarily consists of the independent choice of explanation channels generated by our model and the special co-training procedure. These explanations channels provide a critical advantage in interpretability of regression tasks and a strong boost in predictive performance over previous single-explanation methods. In the following, we will explain in more detail the advantages and novelty we see in our model, and outline our plans to strengthen the manuscript.
>
> **Explanation Channels**
>
> First, we will provide more information on the concept of multi-channel explanations. We will try to fit in as much as possible of that into the revised version of the manuscript (given the strict page limit).
>
> The basic motivation for using multiple channels was the applicability to graph regression problems (one of the most relevant tasks in real-world datasets, e.g. in chemistry and materials science). In that case, we argue that singular attributional explanations (which basically in the best case highlight some sort of edge and node importance) do not provide much useful insight. If a model produces a certain output value “y” and simply marks a certain motif as important, what does that motif actually explain? Does it explain that very specific value, an exceptionally high/low value or is it indicative of some value range? If we do not know that, then single attributional explanations don't provide much value to humans. That is why we introduce multiple explanation channels, where the elements (nodes/edges) of the graph are not only explained by a single [0,1] range importance value, but instead multiple ones, to provide a direct link between explanation channels and prediction values, and thus make the explanations.
>
> In fact, the number of such importance values that are generated for each node/edge of the graph is an architectural hyperparameter of our model. So independent of the task at hand, we can choose the number K of explanations to be generated for each prediction. This number determines the number of attention heads in the convolutional part of the network. So in the end, each attention head is associated with one explanation channel. The attention heads are responsible for the edge explanations. Node importances are then generated by a product of an additional dense importance layer and the pooled edge explanations. We find that exactly this combination of edge and node level explanations is a central part of the working principle of the network as enforces explanations to be connected subgraph-motifs, which again improves the intuitive interpretability of the generated explanations. These [0, 1] node importance masks are then used as the weights in a global pooling operation. To the best of our knowledge, this architectural combination of node- and edge-level importance masks has not previously been explored to orchestrate graph explanations.
>
> Even though any number of explanation channels can be chosen with our architecture, in this work we first explore the most simple choices in more detail, in particular 2 explanation channels for regression tasks. In that way, one explanation channel indicates the parts of the graph that generally supply a positive influence on the label, i.e. that are responsible for high output values. Another explanation channel supplies negative influence, i.e. it shows subgraphs that are responsible for lower target values. We introduce the explanation co-training method to enforce this separation of interpretations of the different channels in a soft and automated manner.
>
> **Reproducibility**
>
> We have provided the hyperparameters for each of the experiments in Appendix B of the paper (due to the strict 9-page limit for the main body of the text). Additionally, the hyperparameters and all implementation details are provided in the source code which as been anonymously published at: https://github.com/awa59kst120df/graph_attention_student. We have failed to mention this fact in a separate Reproducibility statement, but we will do so in the revised version of the manuscript.
>
> **Planned Improvements**
>
> We are working on including more baseline approaches to make the capabilities of our proposed architecture more convincing. Besides other baseline XAI methods we also aim to include comparisons with several state-of-the-art GNN’s, as we have found that our model also achieves surprisingly good results in terms of prediction as well. Furthermore, we want to include an additional real world dataset from material sciences, where we can show that our model is able to reproduce non-trivial structure-property explanations from chemical knowledge, support previously published hypotheses as well as generate new hypotheses for explanatory motifs.

---

> ### Author Response · Authors · 2022-11-18
> **Manuscript Revision**
>
> Dear anonymous Reviewer,
>
> We have uploaded a revised version of our manuscript and provide a summary of the modifications in our official comment: https://openreview.net/forum?id=H6LVUiHzYDE&noteId=p_7u2cPJTr3. We hope, we were able to appropriately address your comments.

---

### Author Response · Authors · 2022-11-18
**Manuscript Revision Summary**

Dear Reviewers,

We would like to thank you again for your comments, constructive criticism and your suggestions for improvements. We acknowledge that the original version of our work failed to clearly communicate the novelty of the approach and was lacking both in sufficient comparison to baseline approaches as well as showcasing real life applicability of our method.

We have extensively revised our manuscript to address your comments and provide a summary of the updated content here:

- First of all, we have adjusted the main scope of our text to point out that the one core novelty of our approach lies within the multi-channel  nature, which is able to provide two independent explanations even for single-value regression tasks. We argue that this is crucial for the interpretability of regression explanations, because only like this it is possible to capture polarity of evidence, where some motifs have a negative influence on the target value and others have a positive influence. Existing single-channel explanation methods, which commonly ignore regression tasks, fail to capture this important information.
- Additionally, we have extended the range of baseline methods used for comparison: In terms of basic procedures we have added simple gradient based explanations and GradCAM explanations in addition to GNNExplainer. More importantly however, we also include the GNES framework of Gao et al. (https://ieeexplore.ieee.org/document/9679041) for GNN explanation supervision. We implement an explanation-supervised GNES model and find that it outperforms all simple baseline procedures with explanation AUC of ~0.85. However, our explanation-supervised model still significantly outperforms all other methods, achieving near perfect explanation AUC of ~0.97. Additionally, we show that when moving to the dual-channel explanations, our model achieves ~0.93 explanation AUC using our explanation co-training routine, even without explanation supervised training.
- We would like to point out that our experimental results are statistically significant, as we perform 50 independent repetitions to obtain the results of almost all computational experiments, except where dataset sizes render this infeasible.
- We have included additional references to other work on self-explaining graph neural networks in our related work section, as pointed in your reviews. However, it is not possible to quantitatively or qualitatively compare to some of them as they focus on explanations types other than the attributional explanations produced in this work.
- In terms of relevance to real-world problems, we apply our model to one additional real-world dataset: The TADF dataset (https://www.nature.com/articles/nmat4717) aims to predict a molecular property called singlet-triplet energy gap which is inportant for organic light emitting diodes. We can show that while achieving good predictivity for target value predictions, our model is also able to reproduce non-trivial structure-property relationships about this task from chemical knowledge. Furthermore, our explanations are able to support previously published hypotheses about possible explanatory motifs and we are even able to propose new hypotheses for explanatory sub-graph motifs for this task.
- Finally, we find that our model shows promising predictive performance beyond the aspect of explainability as well. Therefore we report benchmarking results for several recently proposed GNNs in the appendix, which show that our model matches state-of-the-art predictive performance, ranking second best in two molecular property prediction tasks (solubility datasets ESOL and Lipop). We believe that by including the mixed node and edge level explanation mechanism directly as an integral part of the architecture, the explanation mechanism boosts prediction performance as well.

We sincerely hope that these additions to our work are sufficient to convince you of the novelty and relevance of our proposed method.

---

### Decision · Program_Chairs · 2023-01-20

**Decision:**

Reject

**Justification For Why Not Higher Score:**

Limited technical novelty and the evaluation of explanation on real data is qualitative and anecdotal.

**Justification For Why Not Lower Score:**

N/A

**Metareview: Summary, Strengths And Weaknesses:**

This work proposes  an attention-based self-explaining model for graph regression and classification. The main idea is to multiple explanation channels to produce explanations for graph prediction tasks. The work is specifically motivated by the need for explanations of different polarities for regression problem, hence the need for multiple channels accounting for different polarities.
Experiments are reported on synthetic and real datasets and the proposed method appears to show good performance.
Strength:
Well written paper on an important topic.
experiments on both synthetic and real data produce promising results
Weakness:
The main idea of using multi-channel attention for explanation is straight-forward with limited technical novelty.
Limited experimental results. Authors have included additional baselines based on reviewers comment. That definitely improved the qualify of the evaluation. But the experiments on real data is very cursory. The explanation evaluation is qualitative and anecdotal. The performance comparison was only detailed in the appendix.  As evaluation on real data is a very critical part, the authors should do the due diligence to have a cohesive and complete treatment in the main body of the paper and only defer less important part to the appendix.